# Differentiation of adsorption and degradation in steroid hormone micropollutants removal using electrochemical carbon nanotube membrane

Siqi Liu [1], David Jassby [2], Daniel Mandler [3] & Andrea I. Schäfer [1]✉

The growing concern over micropollutants in aquatic ecosystems motivates the development of electrochemical membrane reactors (EMRs) as a sustainable water treatment solution. Nevertheless, the intricate interplay among adsorption/desorption, electrochemical reactions, and byproduct formation within EMR complicates the understanding of their mechanisms. Herein, the degradation of micropollutants using an EMR equipped with carbon nanotube membrane are investigated, employing isotope-labeled steroid hormone micropollutant. The integration of high-performance liquid chromatography with a flow scintillator analyzer and liquid scintillation counting techniques allows to differentiate hormone removal by concurrent adsorption and degradation. Pre-adsorption of hormone is found not to limit its subsequent degradation, attributed to the rapid adsorption kinetics and effective mass transfer of EMR. This analytical approach facilitates determining the limiting factors affecting the hormone degradation under variable conditions. Increasing the voltage from 0.6 to 1.2 V causes the degradation dynamics to transition from being controlled by electron transfer rates to an adsorption-rate-limited regime. These findings unravels some underlying mechanisms of EMR, providing valuable insights for designing electrochemical strategies for micropollutant control.

Ensuring the safety of the water supply and the health of aquatic environments is an increasingly pressing concern, especially in light of the pervasive presence of endocrine-disrupting chemicals (EDCs), such as steroid hormones (SHs)[1–3]. These substances present considerable risks to both human health and the environment, contributing to psychological disorders, physical health problems, and ecological imbalances, even at trace concentrations[4,5]. The ubiquity of these compounds and the resulting environmental contamination necessitates the urgent need for the development of advanced water and wastewater treatment technologies.

Electrochemical oxidation (EO) is increasingly recognized as a promising approach for sustainable environmental remediation. EO

[1]Institute for Advanced Membrane Technology (IAMT), Karlsruhe Institute of Technology (KIT), Eggenstein-Leopoldshafen, Germany. [2]Department of Civil and Environmental Engineering, University of California, Los Angeles, Los Angeles, CA, USA. [3]Institute of Chemistry, The Hebrew University of Jerusalem, Jerusalem, Israel. ✉e-mail: Andrea.Iris.Schaefer@kit.edu

offers significant advantages over competing technologies, including robust degradation capabilities, no need for continuous chemical supply, versatile and flexible reactor designs, modular construction, and consistent treatment effectiveness due to uniform electric potential distribution[6–8]. EO processes have demonstrated their efficacy in eliminating various micropollutants[9–12], including SHs[13]. Despite these advancements and the tremendous efforts in the development of various electrocatalysts, the full potential of EO often remains underutilized due to the mass transfer limitations of reactants[14].

To address this challenge, recent research breakthroughs have led to the innovation of electrochemical membrane reactors (EMRs) that employ a conducting membrane as a flow-through electrode[7,15–17]. This setup benefits from externally modulated electrochemical potential across the membrane and in its immediate vicinity, thereby facilitates the simultaneous execution of membrane separation and electrochemical treatment. A critical feature of EMRs is the incorporation of nano- to micro-scale networks within the electrode, where the electrochemical reactions are spatially confined within the pores of the EMRs[18–20]. Such a configuration markedly reduces the thickness of the diffusion layer and increases the local concentration of reactants when compared to traditional plate electrodes, resulting in significantly improved mass transfer[21–23]. Furthermore, the efficiency in utilizing active sites within these systems is increased, as the sites are rendered fully accessible to reactant molecules navigating through the microchannels.

Electrochemical membranes transcend traditional membrane functions, extending beyond pure separation to embrace various electro-based strategies via several mechanisms (Fig. 1): i) mass transport of micropollutant to membrane surface, ii) (electro)-adsorption[24], iii) direct electron transfer[25], iv) generation of secondary reactive species[18], and v) desorption of byproducts, which are discussed in detail in the following sections.

Given the significant alleviation of mass transfer limitations through the use of the EMR, the roles of adsorption/desorption and electron transfer processes have become increasingly crucial in the degradation of micropollutants. Decades of research underpin the profound connection between the adsorption characteristics of reactants and their intermediates on electrode surfaces and the overall kinetics of electrochemical reactions. Within an electrochemical system characterized by specific electrodes, electrolytes, reactants, and temperature conditions, the kinetics of reactions involving heterogeneous electron transfer are significantly influenced by the surface coverage of the reactants[26,27]. This dynamic is largely governed by the local concentration of the reactant near the electrode surface and the adsorption constant[28], or the affinity of the reactant for the material. Applying an electrochemical potential can exert a notable influence on adsorption, mediated by dipole and electrostatic interactions between the electrode and reactants[24], and/or competition from other adsorbed species. Zheng et al.[29] further elucidate this principle, demonstrating, for instance, the profound effects that variations of the electrochemical potential have on the adsorption behavior of CO at Ru films, thereby influencing the electrochemical pathway and the nature of the resultant products.

Upon molecular adsorption at the electrode surface, the application of sufficient potential initiates heterogeneous electron transfer, catalyzing the electrochemical transformation of micropollutants[30]. The kinetics of these reactions are primarily governed by the inner-sphere electron transfer mechanism, wherein the reactant species typically involve a strong interaction with the electrode surface[31]. This mechanism is significantly influenced by the reactive site density, surface structure, and chemical composition of the electrode[32].

In many electrochemical systems[18,21,33], the in situ electrochemical generation of reactive species, such as the hydroxyl radical ($\cdot$OH)[34,35], at and near the electrode surface is a dominant mechanism for degrading micropollutants. The ubiquity of chloride ions (Cl$^-$) in natural waters and wastewater effluents often leads to the formation of active species in electrochemical processes[36]. Oxygen reduction on the cathode, through a two-electron process in acidic or neutral media, produces hydrogen peroxide ($H_2O_2$), another potent oxidizer of certain organic compounds[37].

Recent advances in water treatment with the EMR process have focused on developing sophisticated conducting membrane materials based on carbonaceous materials[38], metals and metal oxides[39], as well as polymers[40]. Among these, carbon nanotubes (CNTs), owing to their array of unique physical and chemical attributes[41,42], have emerged as particularly prominent, and have been extensively studied for their application in EM fabrication[43]. CNTs, with a graphene-like structure, demonstrate high conductivity (multi-walled CNTs exhibiting a range of 1000 to 200,000 S cm$^{-1}$[44–46]), facilitating efficient electron transfer. Moreover, the presence of topological defects on the CNT surface is believed to further enhance this electroactivity[47]. The distinctive nano-scale tubular structure of CNTs endows with an immense surface area (50 to 1000 m$^2$ g$^{-1}$[48]), which provides abundantly accessible surface sites for effective adsorption of diverse organic compounds, thereby

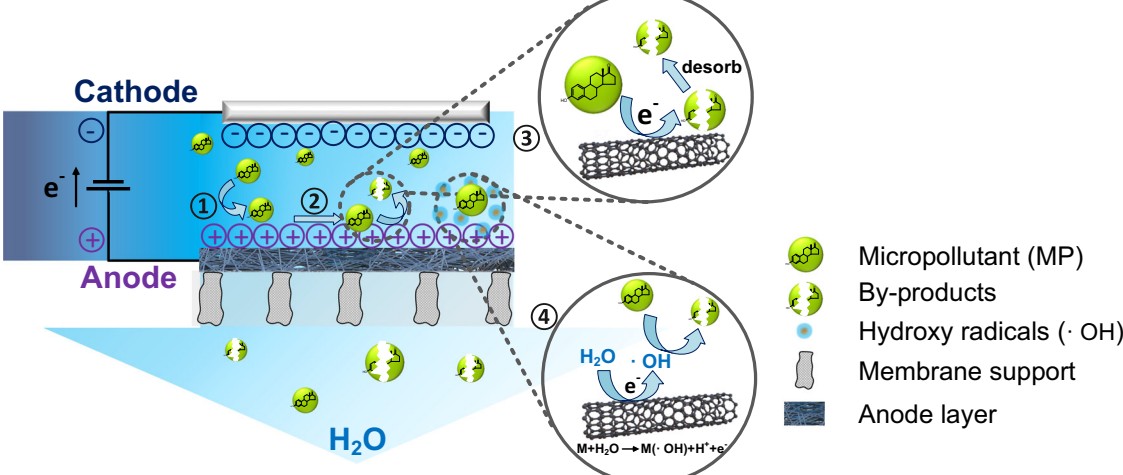

**Fig. 1 | Illustration of mechanism.** Schematic diagram of the mechanisms of an electrochemical membrane reactor (EMR) with cathode above the membrane anode. ① Mass transfer of MP to electrochemical membrane surface; ② Adsorption of MP on electrochemical membrane surface; ③ Direct electron transfer between MP and electrochemical membrane; ④ Reaction with secondary reactive species in bulk solution (occurring along with step ③).

facilitating subsequent electrochemical reactions[49]. Furthermore, leveraging the high aspect ratio and mechanical strength, CNTs can be easily formed into porous three-dimensional networks with high porosity at concentrations as low as 0.01 wt%[50]. CNTs, which typically exhibit a low overpotential for $O_2$ evolution (generally <0.4 V[51]), are considered as active anodes (M). These materials interact strongly with electrogenerated ·OH to form a higher-state oxide (MO), $M(\cdot OH) \rightarrow MO + H^+ + e^-$, that in combination with the anode surface, acts as a selective mediator in degrading organic compounds via direct oxidation[52,53].

Despite the great potential of EMRs in micropollutant degradation, the operational mechanisms are not fully understood, primarily due to the intricate interplay of adsorption/desorption dynamics, electrochemical reactions, and byproduct formation. This lack of clarity arises from the inability of current analytical tools to quantify these processes in complex environments and limits process design. To overcome these challenges, this work employed isotope-labeled SH micropollutants as reporter compounds, and a CNT electrochemical membrane as a flow-through anode.

The goal of this study is to unravel the underlying mechanisms of EMR by employing a combination of high-performance liquid chromatography-flow scintillator analyzer (HPLC-FSA) and liquid scintillation counting (LSC) techniques, offering a more comprehensive understanding of the sophisticated EMR system, and address the following research questions; i) Can the simultaneous adsorption and desorption of isotope-labeled micropollutant be quantified in the EMR? ii) How does pre-adsorption limit the removal of SH micropollutants in EMR? iii) To what extent are electrochemical adsorption, degradation and formation of byproducts determined by the system conditions, including cell voltage, water flux, SH concentration, and SH types?

## Results and Discussion

An innovative approach employing isotopically labeled SHs was adopted to investigate the removal mechanisms of SH micropollutants in a CNTs EMR. This method focused on quantifying the simultaneous processes of electrochemical adsorption and degradation, elucidating their contributions to SH removal under various operational and solution conditions.

### Adsorption and degradation of β-estradiol in the EMR

The removal of E2 in the CNT EMR in the flow-through system is shown in Fig. 2, where the samples were analyzed through a combined approach of UHPLC-FSA and LSC to quantify the concurrent adsorption and degradation process occurring within the CNT membranes.

Both the normalized $^3H$ activity ($c_{p,3H}/c_{f,3H}$) and normalized E2 concentration ($c_{p,E2}/c_{f,E2}$) in permeate exhibited a consistent increase to $0.6 \pm 0.09$ ($40 \pm 9\%$ removal) across an accumulative permeate volume of 500 mL, due to the adsorption of E2 during a pass-through the membrane. Saturation of the membrane with E2 was not reached prior to applying cell voltage. The application of 1.6 V voltage caused an immediate surge in the normalized $^3H$ activity, to $4.3 \pm 0.14$. This increase could be caused by the electrochemical desorption of E2 that was pre-adsorbed onto the membrane due to the competition on the part of anionic species[54], such as $Cl^-$ and $HCO_3^-$ present in the electrolyte, and possibly also due to the desorption of its degradation products, triggered by the applied voltage. Subsequently, the normalized $^3H$ activity exhibited a declining trend, eventually stabilizing as the permeate volume increased. This pattern suggested that an equilibrium was reached, where the desorption rate of the radiolabeled compounds matched the adsorption rate.

After activating the electric power, the E2 concentration consistently decreased with the permeate volume, eventually descending below the detection limit of 2.5 ng L$^{-1}$. This implied that the discharge of $^3H$-labeled compounds from the membrane originated predominantly from the degradation products of E2. The degradation performance of E2 achieved within the CNT EMR outperformed that of a comparable photocatalytic membrane reactor, which exhibited about 80% E2 removal under the same flux condition with 10 mW cm$^{-2}$ UV irradiation, despite using a smaller membrane surface area (2 cm$^2$) and a lower initial E2 concentration (100 ng L$^{-1}$)[55]. This superior performance made the EMR a promising option for micropollutant remediation.

While LSC could determine the $^3H$, it cannot differentiate between intact and degraded E2. This can be differentiated using UHPLC-FSA. Employing the methodology specified in Eqs. (2–5), the total mass of E2 eliminated in the EMR constituted roughly $76 \pm 10\%$ of its initial amount in the feed. Of this, approximately $61 \pm 7\%$ of E2 was converted into byproducts and subsequently transported into the permeate, whereas the remaining $15 \pm 3\%$ was retained on the membrane, comprising either E2 and/or its byproducts. Conversely, about $24 \pm 10\%$ of the E2 remained in the permeate. This residual E2, constituting $19 \pm 4\%$ of the total, was ascribed to the fraction that did not undergo adsorption before activating the voltage. Details of the mass balance analysis can be found in SI (Supplementary Fig. 24A).

The mass balance analysis conducted across different permeate volume ranges revealed a clear trend: the mass removal by adsorption tended to stabilize in comparison to the increase in degradation as permeate volume increased (Supplementary Fig. 1). Upon activating the voltage at 500 mL of permeate, the total removed mass of E2

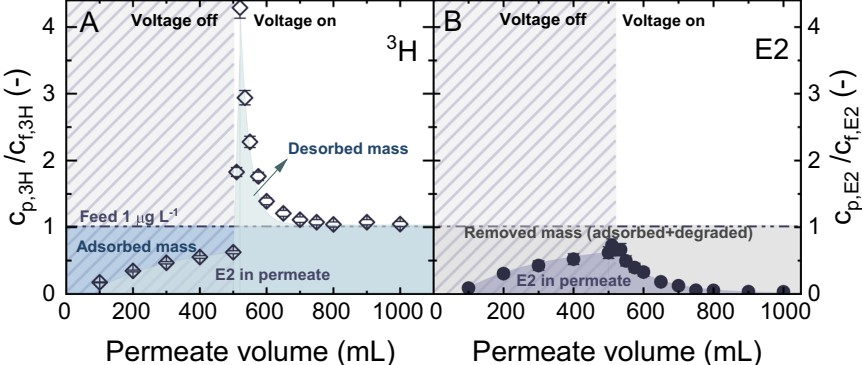

**Fig. 2 | Electrochemical adsorption and degradation of estradiol (E2).**
**A** Normalized total tritium ($^3H$) activity ($c_{p,3H}/c_{f,3H}$) measured via LSC, and (**B**) normalized concentration of E2 ($c_{p,E2}/c_{f,E2}$) measured via UHPLC-FSA in permeate vs. accumulated permeate volume. $c_{f,E2} = 1\,\mu g\,L^{-1}$, $V_{cell} = 1.6\,V$, $J_f = 150\,L\,m^{-2}\,h^{-1}$ (5 mL min$^{-1}$), 1 mM NaHCO$_3$, 10 mM NaCl, 27.2 mg L$^{-1}$ EtOH, 79.2 mg L$^{-1}$ MeOH, pH 8.2 ± 0.2, 23 ± 1 °C. Error bars represent propagated error from operational parameter variations and analytical error.

increased from $63 \pm 1\%$ to $76 \pm 10\%$ as the permeate volume rose from 500 to 1000 mL. Within this, the contribution from degradation consistently increased from 0 to $61 \pm 7\%$. The adsorption contribution decreased from $63 \pm 1\%$ to $19 \pm 3\%$ with the permeate volume increase from 500 to 800 mL and then plateaued in the range of 15–19% upon further increasing the permeate volume to 1000 mL. Notably, after 800 mL of filtration volume, both the normalized $^3H$ activity and E2 concentration stabilized at 1 and 0, respectively (Fig. 2), indicating an equilibrium state where all incoming E2 molecules were degraded and the formed byproducts penetrated the permeate. However, a small fraction of E2 remained adsorbed on the membrane after 800 mL of permeate volume, suggesting some sites on the CNT surface were ineffective or unable to initiate the electrochemical reactions. The possible reasons will be further discussed in the subsequent section. These results suggested that an increase in total mass removal is anticipated with extended continuous operation (not accounting for the stability of the membrane), though a small portion of the micropollutants may remain adsorbed on the membrane.

## Byproduct formation along with degradation of β-estradiol

While the CNT EMR process demonstrated effective degradation of the target micropollutant, complete mineralization could not be achieved under the conditions tested. This partial degradation, leading to the formation of byproducts, identified by the UHPLC-FSA chromatogram analysis (Fig. 3).

Figure 3A shows the emergence of two distinct peaks located at around 3 and 8 minute retention times in the UHPLC-FSA chromatograms for the sample collected at 520 mL. Even though the exact chemical compositions cannot be identified, these peaks were clearly associated with the byproducts generated through the electrochemical transformation of E2. These peaks were designated as byproduct-3m and byproduct-8m, respectively. The release of these byproducts into the permeate facilitated the regeneration of adsorptive sites on the membrane surface, thereby enabling the adsorption of incoming E2 and its subsequent oxidation. This dynamic process contributed to the observed decline in the E2 peak, accompanied by an increase in the peaks of byproducts at 520 mL.

Prior studies[56–58] have identified a ketone derivative as a product of the direct electrochemical oxidation of E2 through a two-electron transfer mechanism, as evidenced by GC-MS analysis. Such a transformation was anticipated to weaken the estrogenic activity of E2. This effect was primarily because of the conversion of the phenol group, a crucial structural component that enables endocrine-disrupting chemicals to exhibit their estrogenic properties through interaction with the human estrogen receptor, into a quinone-like moiety[57,59,60]. Further investigation into the estrogenic activity of the EMR-treated SH water is crucial and is currently in progress. It is posited that the ketone

derivative corresponds to the peak observed at 8 minutes in the UHPLC-FSA chromatograms. This derivative would subsequently undergo additional oxidation, yielding a range of cleaved byproducts that were eluted between 2 to 4 minutes in the UHPLC analysis.

Figure 3B shows that the sum of the normalized concentration of E2, and its two byproducts ($c_{p,E2+prod}/c_{f,E2+prod}$) closely aligned with the normalized $^3H$ activity ($c_{p,3H}/c_{f,3H}$). This implies that byproduct-3m and byproduct-8m constituted the primary byproducts of E2 degradation, with no evidence indicating the formation of other undetectable byproducts.

## Role of pre-adsorption

The adsorption of reactants on the electrode interface is a prerequisite for the subsequent heterogeneous electron transfer reaction[61]. The pre-adsorption of the micropollutant onto the EMR interface prior to voltage application is believed to enhance its electrochemical degradation. However, investigation into the role of pre-adsorption has been limited by the absence of analytical methods capable of distinctly differentiating between adsorption and degradation processes in electrochemical settings. The fact that CNT membranes provide a vast surface area, combined with the extremely low concentrations at which micropollutants occur, makes achieving saturation difficult. To elucidate the role of pre-adsorption, an experiment was carried out without pre-adsorption (Fig. 4).

Following the application of voltage at the start of filtration, the E2 concentration in the permeate remained below the detection limit throughout the 500 mL process run, while normalized $^3H$ activity surged from 0 to $0.9 \pm 0.3$ ($10 \pm 3\%$ removal) before stabilizing (Fig. 4B). This rapid increase in the normalized $^3H$ activity implied a predominance of the electrochemical degradation over adsorption in the EMR process, evidenced by byproduct formation accompanied by the E2 removal (Fig. 4A).

Comparison of the E2 degradation with pre-adsorption (Fig. 4C) showed that, without pre-adsorption, a total of $98 \pm 8\%$ of the initial E2 mass was eliminated within the EMR, while that in the case of with pre-adsorption is $76 \pm 10\%$. This 22% difference could be mainly attributed to the fraction of E2 being released into the permeate prior to the application of voltage ($19 \pm 4\%$). Additionally, after turning on the voltage, $5 \pm 2\%$ of the E2 was released into the permeate, which was comparable to the $2 \pm 2\%$ of E2 that remained unremoved without pre-adsorption. Details of the mass balance analysis for the experiments with and without pre-adsorption are available in Supplementary Fig. 24. Among the $98 \pm 8\%$ of the removed E2 mass without pre-adsorption, $77 \pm 3\%$ was removed through electrochemical degradation, and $22 \pm 4\%$ remained adsorbed on the membrane. These findings indicated that pre-adsorption of E2 did not significantly impact, either positively or negatively, the subsequent removal process. This can be

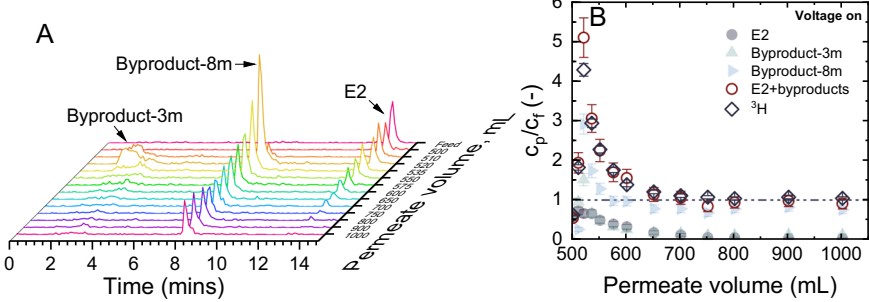

**Fig. 3 | Formation of byproduct during degradation. A** UHPLC-FSA chromatograms of estradiol (E2) during the electrochemical degradation phase (after 500 mL) with increasing accumulated permeate volume; **B** normalized concentration of E2 ($c_{p,E2}/c_{f,E2}$), byproduct-3m ($c_{p,prod3}/c_{f,prod3}$), byproduct-8m ($c_{p,prod8}/c_{f,prod8}$), and the sum of E2 and the two byproducts at 3 and 8 min

($c_{p,E2+prod}/c_{f,E2+prod}$), and normalized total tritium ($^3H$) activity ($c_{p,3H}/c_{f,3H}$) vs. accumulated permeate volume from 500 to 1000 mL. $c_{f,E2} = 1\,\mu g\,L^{-1}$, $V_{cell} = 1.6\,V$, $J_f = 150\,L\,m^{-2}\,h^{-1}$ (5 mL min$^{-1}$), 1 mM NaHCO$_3$, 10 mM NaCl, 27.2 mg L$^{-1}$ EtOH, 79.2 mg L$^{-1}$ MeOH, pH $8.2 \pm 0.2$, $23 \pm 1$ °C. Error bars represent propagated error from operational parameter variations and analytical error.

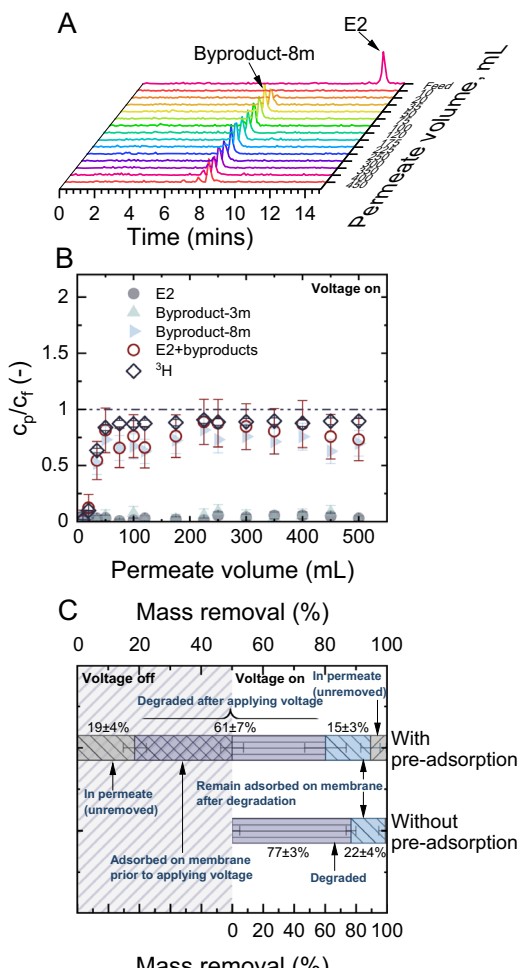

**Fig. 4 | Role of pre-adsorption in the micropollutant removal. A** UHPLC-FSA chromatograms of estradiol (E2) during the electrochemical degradation phase over permeate volume; **B** normalized concentration of E2 ($c_{p,E2}/c_{f,E2}$), byproduct-3m, byproduct-8m, the sum of E2 and the two byproducts, and normalized total tritium ($^3$H) activity *over* permeate volume from 0 to 500 mL; and **C** contribution to the mass removal of E2 by the electrochemical adsorption and degradation. $c_{f,E2} = 1\,\mu g\,L^{-1}$, $V_{cell} = 1.6\,V$, $J_f = 150\,L\,m^{-2}\,h^{-1}$ (5 mL min$^{-1}$), 1 mM NaHCO$_3$, 10 mM NaCl, 27.2 mg L$^{-1}$ EtOH, 79.2 mg L$^{-1}$ MeOH, pH 8.2 ± 0.2, 23 ± 1 °C. Error bars represent propagated error from operational parameter variations and analytical error.

attributed to; i) the apparent adsorption rate not being a limiting factor for E2 degradation in the CNT EMR under these operational conditions, owing to the high adsorption capacity of the CNT membrane and the elimination of mass transfer limitations within EMR, and ii) the desorption rate of the byproducts of E2 being rapid enough, which did not hinder the continuous adsorption of incoming E2.

In contrast to experiments with pre-adsorption, byproduct-3m was not detected in the HPLC-FSA chromatograms (Fig. 4A). This suggested that the electrochemical mineralization of E2 within the CNT EMR was primarily governed by the rate of transformation from byproduct-8m to byproduct-3m. Notably, this transformation rate was slower than the rate at which byproduct-8m forms from E2. The pre-adsorption of E2 significantly facilitated the rapid production of byproduct-8m upon activating the voltage. This resulted in the formation of a substantial amount of byproduct-8m, which was then converted into byproduct-3m, as indicated by the prominent peak between 2-4 min in the early phase of voltage application (Fig. 3A). As the majority of the pre-adsorbed E2 underwent degradation, the peak for byproduct-3m vanished after 650 mL of permeate had passed

through. Without pre-adsorption, an insufficient quantity of byproduct-8m accumulated on the membrane, resulting in a quantity of byproduct-3m too small to produce a prominent peak in the chromatogram.

These findings demonstrated the significance of the integrated UHPLC-FSA and LSC analytical method to elucidate the intricate processes occurring within the EMR. By employing this novel analytical approach, the limiting factors of the EMR can now be determined across variable operational conditions (cell voltage, flux, concentration, SH types), thereby offering profound insights into the underlying mechanisms of EMR.

## Contribution of electron transfer rate to removal

Previous sections established that pre-adsorption does not limit the electrochemical degradation of E2 within the CNT EMR. The contribution of electron transfer rate at the electrode surface with pre-adsorption, which is modulated by the applied cell voltage, was examined in a voltage range from 0.6 to 2.5 V (Fig. 5).

The evolution of UHPLC-FSA chromatograms, and the profile of normalized permeate concentration of E2 ($c_{p,E2}/c_{f,E2}$), formed byproducts ($c_{p,prod}/c_{f,prod}$), and the $^3$H activity ($c_{p,3H}/c_{f,3H}$) *vs.* the permeate volume for all parameters can be found in the SI, as well as the system parameters (conductivity, pH, temperature, and transmembrane pressure) recorded during the experiments.

By applying a cell voltage of 0.6 V, a steady-state removal of 35 ± 8% of E2 was achieved, which then increased rapidly to 97 ± 2% with increasing the cell voltage to 1.2 V. Subsequently, E2 removal plateaued as further increasing the voltage to 2.5 V (Fig. 5A).

Mass balance calculations showed that 55 ± 9% of the total E2 mass was eliminated from the system at 0.6 V, with 41 ± 2% attributed to adsorption and 14 ± 7% to electrochemical degradation (Fig. 5B). Additionally, UHPLC chromatograms (Supplementary Fig. 3) identified minor formation of byproducts at 0.6 V. Increasing the voltage to 1.2 V resulted in a rise in the total E2 mass removal from 55 ± 9 to 79 ± 9%. With this increase, the contribution of adsorption decreased from 41 ± 2% to 22 ± 3%, signifying a shift from adsorption to degradation with increasing cell voltage. Alongside the two byproducts-3m and -8m, the emergence of an additional byproduct, approximately at 5 minutes (denoted as byproduct-5m) was observed in the UHPLC-FSA chromatograms at 0.9 and 1.2 V. Beyond the 1.2 V threshold, the amount of E2 removed, along with the contributions from adsorption and degradation, plateaued. The peak intensity of byproduct-5m consistently declined as voltages exceeding 1.2 V and disappeared entirely at 1.6 V. This suggested that the electron transfer rate of the byproduct-5m at the CNT surface was significantly higher than those of byproducts -3m and -8m.

As discussed in Fig. 2, the mass balance calculations indicated that the E2 remaining in the permeate primarily originated from the adsorption phase. This suggested that nearly all E2 molecules subjected to the degradation phase were removed within the EMR at voltages > 1.2 V, through either adsorption or electrochemical transformation. Notably, the proportion of E2 removed through degradation became independent of cell voltage > 1.2 V, implying that cell voltage ceased to be a limiting factor above this threshold. Given that all E2 molecules were adsorbed onto the membrane (with E2 concentration in the permeate being <2.5 ng L$^{-1}$) during the degradation phase, the apparent adsorption rate was not deemed to be the limiting factor for the subsequent degradation of E2. Another potentially limiting factor was speculated to be the non-uniform distribution of electroactive sites on the membrane surface. CNTs lack a perfect surface with sp$^2$-hybridized structure and are characterized by numerous defects[47]. Previous research has indicated that electroactive sites are predominantly located at these defects[62–64]. Consequently, the E2 molecules might adsorb onto the less electroactive sites, potentially resulting in slower rates or an inability for subsequent degradation. Although prior studies have

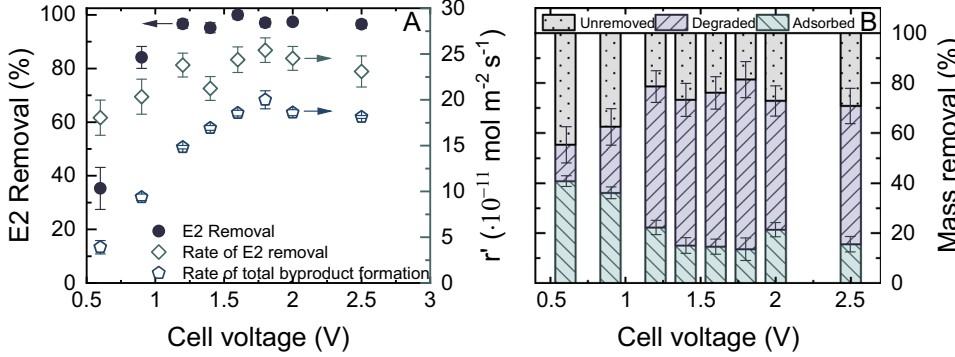

**Fig. 5 | Impact of cell voltage on the micropollutant removal.** Electrochemical degradation of estradiol (E2) over cell voltage from 0.6 to 2.5 V, expressed as (**A**) E2 removal, apparent rate of E2 removal, and apparent rate of total byproducts formation; and (**B**) contribution to the mass removal of E2 by the electrochemical adsorption and degradation. $c_{f,E2}$ = 1 µg L$^{-1}$, V, $J_f$ = 150 L m$^{-2}$ h$^{-1}$ (5 mL min$^{-1}$), 1 mM NaHCO$_3$, 10 mM NaCl, 27.2 mg L$^{-1}$ EtOH, 79.2 mg L$^{-1}$ MeOH, pH 8.2 ± 0.2, 23 ± 1 °C. Error bars represent propagated error from operational parameter variations and analytical error.

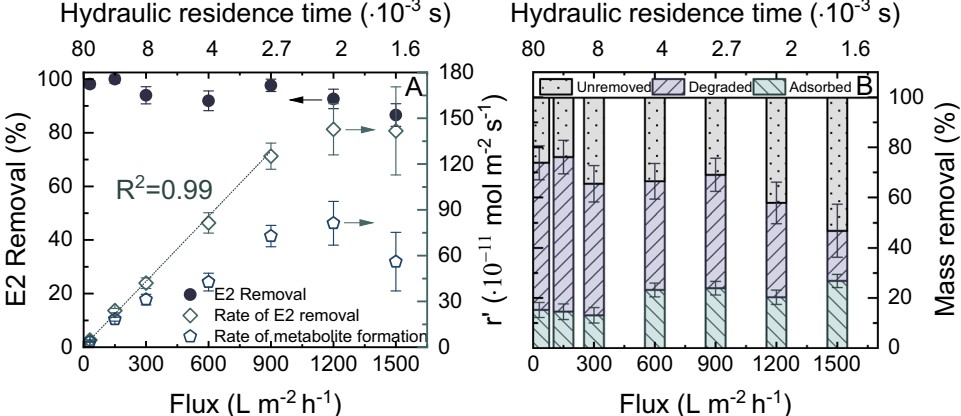

**Fig. 6 | Impact of flux on the micropollutant removal.** Electrochemical degradation of estradiol (E2) over flux from 60 to 1500 L m$^{-2}$ h$^{-1}$, expressed as (**A**) E2 removal, apparent rate of E2 removal, and apparent rate of total byproducts formation; and (**B**) contribution to the mass removal of E2 by the electrochemical adsorption and degradation. $c_{f,E2}$ = 1 µg L$^{-1}$, V, $V_{cell}$ = 1.6 V, 1 mM NaHCO$_3$, 10 mM NaCl, 27.2 mg L$^{-1}$ EtOH, 79.2 mg L$^{-1}$ MeOH, pH 8.2 ± 0.2, 23 ± 1 °C. Error bars represent propagated error from operational parameter variations and analytical error.

emphasized the importance of topological defects on the CNT surface in electrochemical applications[65], this aspect is often overlooked in the EMR process. This area warrants further investigation using advanced technologies (such as magnified scanning tunneling microscopy[66] and density functional theory calculations[67]) not available in the current scope of work.

These results demonstrated a fixed portion of E2 molecules encountered difficulties in undergoing electrochemical transformation and desorption from the membrane unaffected by the adsorption kinetics and electron transfer rate. To address this aspect of SH treatment in the EMR, the flux was varied to determine whether a hydraulic residence time ($t_r$) is limiting the degradation by reducing the time available for E2 molecules adsorbing onto the membrane and undergoing degradation.

### Dependence of removal on hydraulic residence time

Flux is the flow rate normalized by membrane area and this determines the $t_r$ within the EMR, thereby determining the duration of contact between the micropollutant and membrane surface. This also directly impacts the quantity of micropollutant molecules brought into the membrane per unit of time.

Water flux on the E2 removal within the CNT EMR was studied at varying fluxes from 30 to 1500 L m$^{-2}$ h$^{-1}$ (Fig. 6). Correspondingly, the

hydraulic residence time within this flux range decreased from 80·10$^{-3}$ to 1.6·10$^{-3}$ s.

Increasing the flux from 30 to 1500 L m$^{-2}$ h$^{-1}$ did not significantly influence the E2 removal, which was varied between 87 ± 4 and 99 ± 2%. The apparent rate of E2 removal increased linearly ($R^2$ = 0.99), from (4.7 ± 0.03)·10$^{11}$ to (125 ± 9)·10$^{11}$ mol m$^{-2}$ s$^{-1}$ in the flux range of 30 to 900 L m$^{-2}$ h$^{-1}$ ($t_r$ reduced from 80·10$^{-3}$ to 2.7·10$^{-3}$ s). Beyond this flux, the apparent rate of removal plateaued at about (142 ± 29)·10$^{11}$ mol m$^{-2}$ s$^{-1}$. This suggested that the kinetics of E2 removal was initially constrained by the availability of E2 molecules within the EMR per unit time, particularly when $t_r$ > 2.7·10$^{-3}$ s. With increase in flux, more E2 molecules were available. At higher flux, the E2 removal rate may be limited by either inadequate contact time for reactions or insufficient sites for adsorption. This necessitates a more detailed analysis of the mass balance within the EMR to unravel the underlying mechanisms.

The fraction of the E2 mass removed in the EMR decreased consistently from 74 ± 10 to 47 ± 13%, accompanied with a reduction in the degradation contribution from 59 ± 7 to 20 ± 11% and an increment in the adsorption contribution from 15 ± 3 to 27 ± 3% (Fig. 6B). These findings implied that the influence of contact time on electrochemical reactions is notably more pronounced than that on the adsorption/desorption dynamics. Higher flux provided insufficient duration for the E2 molecules being degraded at the CNT surface.

Analysis of the UHPLC-FSA chromatograms (Supplementary Fig. 6) revealed the presence of a peak for byproduct-5m when flux exceeding 1500 L m$^{-2}$ h$^{-1}$. This observation suggested that the transformation of byproduct-5m was constrained by insufficient contact time at higher fluxes, highlighting the necessity for extended residence time to facilitate the elimination of byproduct-5m.

The preceding discussion proved the efficacy of the CNT EMR in eliminating E2 at a relatively low concentration (1 μg L$^{-1}$), unimpeded by limitations from the electron transfer rate (at cell voltage of 1.6 V) or contact time (at water flux of 150 L m$^{-2}$ h$^{-1}$ and average 0.7 bar). Compared to nanofiltration, which typically operates at fluxes ranging from 4.5 to 600 L m$^{-2}$ h$^{-1}$ under pressures of 3-20 bar[68], the EMR process offers significant advantages due to its lower energy consumption and the elimination of concentrate production. The extent to which the E2 quantity available in the membrane pores was then explored at these operational conditions though investigation of concentrations.

## Degradation dependence on β-estradiol concentration

Electrochemical removal of E2 within the CNT EMR across a wide array of E2 concentrations (50 to 10$^6$ ng L$^{-1}$) is shown in Fig. 7. This comprehensive analysis aimed to ascertain the maximal concentration threshold of E2 that the EMR can effectively process at the specified experimental conditions.

The steady-state E2 removal consistently exceeded 95 ± 3% for the concentrations ranging from 50 to 5·10$^5$ ng L$^{-1}$ but a sharp decline to 33 ± 10% occurs when the E2 concentration is increased to 10$^6$ ng L$^{-1}$ (Fig. 7A). The correlation between the rate of E2 removal and its concentration displayed a linear progression up to 5·10$^5$ ng L$^{-1}$ ($R^2$ = 0.99), then plateaued at 10$^6$ ng L$^{-1}$. A trend mirrored by the rate of byproduct formation. This pattern indicated that, at an E2 concentration of 10$^6$ ng L$^{-1}$, the quantity of E2 in the EMR surpasses the amount that can be adsorbed and degraded.

While the steady-state E2 removal in the range of 50 to 5·10$^5$ ng L$^{-1}$ remained stable, the proportion of E2 mass removed from the initial feed consistently decreased from 88 ± 10 to 37 ± 12% within this concentration range (Fig. 7B), which is primarily attributed to the fraction of E2 that bypassed the system prior to applying the voltage. At a higher concentration, both adsorption and degradation contributions decreased concurrently. Beyond 10$^6$ ng L$^{-1}$, the total removed E2 dropped further to 17 ± 14%. Remarkably, the contribution of degradation was reduced from 35 ± 9 to 12 ± 10%, implying that at concentrations > 5·10$^5$ ng L$^{-1}$, the apparent rate of electrochemical degradation of E2 became governed by the electron transfer rate. The adsorption contribution declined sharply at concentrations higher

than 10$^4$ ng L$^{-1}$. It is worth noting that the mass of adsorbed E2 consistently increased with the rising E2 concentrations. However, due to the large initial mass of E2, the proportion of adsorbed E2 appears negligible at concentrations > 10$^4$ ng L$^{-1}$.

Upon identifying the operational parameters that restrict E2 removal within the EMR, SHs with varying molecular structures were examined to determine which chemical bonds influence electrochemical reactions at the CNT surface.

## Degradation of different types of steroid hormones

Four types of SH, namely E1, E2, T, and P, were selected to investigate whether the molecular structures affect degradability (Fig. 8). Both E1 and E2 feature an aromatic ring, whereas neither P nor T features an aromatic ring. E2 is characterized by two hydroxyl groups, compared to E1 that contains only a single hydroxyl group.

Prior to the application of voltage, the normalized concentrations of E1, E2 and P in the permeate increased to 0.75 ± 13 0.6 ± 0.09 and 0.6 ± 0.12, respectively, whereas the normalized T concentration increased to 0.95 ± 0.15. This indicated that the CNT membrane had a lower adsorption capacity for T compared to other SH types. Upon applying a voltage of 1.6 V, a normalized permeate concentration 0.7 ± 0.12 (30 ± 12% removal) was observed for T at the end of the experiment (1000 mL), while P showed a concentration ratio around 1 (no significant removal). The normalized $^3$H activity stabilized at around 1 after 800 mL of permeate volume. These findings indicated a markedly lower electrooxidative activity for T and P in comparison to E2. The observed challenges in oxidizing T and P are believed to have arisen from the presence of oxidation-resistant quinone structures in their molecular structures[69,70].

At the end of the experiment, the normalized concentration of E1 decreased to 0.12 ± 0.04 (88 ± 4% removal). Notably, the $^3$H activity also decreased to 0.37 ± 0.15, suggesting a significant contribution from adsorption to the removal of E1. Mass balance analysis revealed that 54 ± 2% of the initial E1 mass was removed through adsorption, while only 2 ± 9% underwent degradation. This finding was supported by the UHPLC-FSA chromatograms (Fig. S19), which showed no distinct byproducts formed during the experiment. The decrease in $^3$H activity could be explained by the possibility that the degradation products of E1 differed from those of E2, remaining adsorbed on the membrane with minimal release into the permeate, but this could not be validated.

These results demonstrated that the electrochemical removal of micropollutants is highly dependent on the molecular structures that influence both the adsorption kinetics and degradability. Although E1 and E2 have very similar structures, their degradation behaviors within

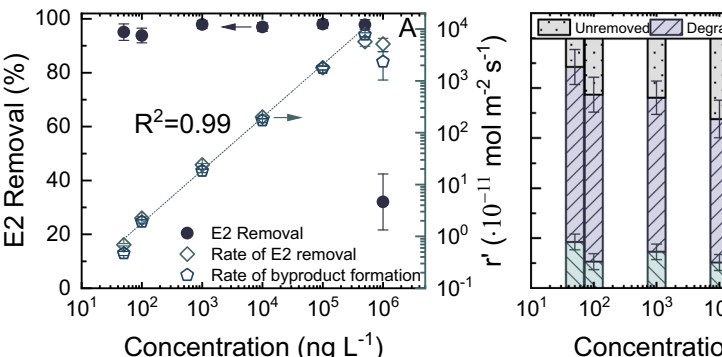

**Fig. 7 | Impact of feed concentration on the micropollutant removal.** Electrochemical degradation of estradiol (E2) over concentration from 50 to 10$^6$ ng L$^{-1}$, expressed as (**A**) E2 removal, apparent rate of E2 removal, and apparent rate of total byproducts formation; and (**B**) contribution to the mass removal of E2 by the electrochemical adsorption and degradation. $V_{cell}$ = 1.6 V, $J_f$ = 50 L m$^{-2}$ h$^{-1}$ (5 mL min$^{-1}$), 1 mM NaHCO$_3$, 10 mM NaCl, 27.2 mg L$^{-1}$ EtOH, 79.2 mg L$^{-1}$ MeOH, pH 8.2 ± 0.2, 23 ± 1 °C. Error bars represent propagated error from operational parameter variations and analytical error.

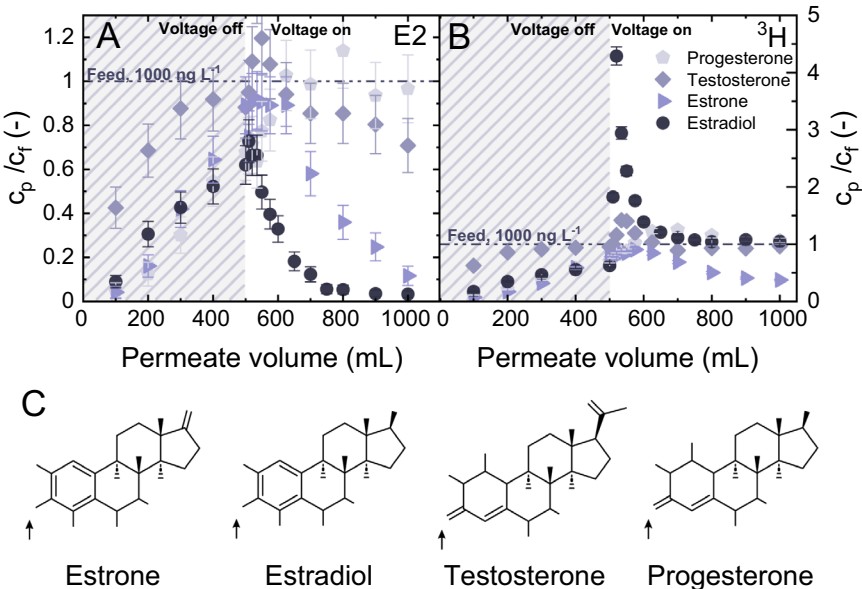

**Fig. 8 | Impact of hormone types on the micropollutant removal.** Electrochemical degradation of individual steroid hormones (SHs), expressed as normalized concentration of (**A**) SH, and (**B**) [3]H *vs.* accumulated permeate volume; (**C**) Chemical structure of the four SH types used in this study.

$c_{f,SH} = 1 \, \mu g \, L^{-1}$, $V_{cell} = 1.6 \, V$, $J_f = 150 \, L \, m^{-2} \, h^{-1}$ (5 mL min⁻¹), 1 mM NaHCO₃, [1]0 mM NaCl, 27.2 mg L⁻¹ EtOH, 79.2 mg L⁻¹ MeOH, pH 8.2 ± 0.2, 23 ± 1 °C. Error bars represent propagated error from operational parameter variations and analytical error.

the CNT EMR differ markedly. Increased adsorption capacity did not necessarily compensate for low degradability. For example, despite P showing greater adsorption on the membrane compared to T, T exhibited higher degradation efficiency.

In summary, an investigation into the electrochemical degradation of isotope-labeled SHs in a flow-through, single-pass EMR equipped with a composite microfiltration membrane with a CNT conductive layer has been performed under various operational and solution conditions. Both UHPLC-FSA with LSC was applied to differentiate the contribution of electrochemical adsorption and degradation in the SH removal, allowing unraveling the underlying mechanisms of the CNT EMR.

It was found that, 78 ± 10% of E2 mass in feed was removed, among which approximately 60 ± 7% of the initial E2 was removed through electrochemical degradation, as identified by the combined UHPCL-FSA and LSC analytical tools. An E2 permeate concentration below the LOD of 2.5 ng L⁻¹ was achieved, treating an influent SH concentration of 1 µg L⁻¹ with a cell voltage of 1.6 V, and a flux of 150 L m⁻² h⁻¹. The pre-adsorption of E2 onto the membrane before voltage application did not alter the degradation. This was attributed to the high adsorption capacity of the CNT membrane and the fast reaction kinetics in degradation. An increase in cell voltage from 0.6 to 1.2 V consistently enhanced the proportion of electrochemical transformation to total mass removal, reaching a plateau with further voltage increasing to 2.5 V, this indicated that the electron-transfer rate at the membrane surface is a limiting factor for E2 degradation at voltages below 1.2 V. Approximately 20% of E2 was found to be removed via adsorption on the membrane surface without undergoing degradation, regardless of voltages exceeding 1.2 V. This undegraded E2 was ascribed to its adsorption onto non-electroactive sites. The results identified electroactive sites as a crucial, yet often overlooked, limiting factor in CNT EMR studies. An extended residence time allowed for a prolonged interaction between E2 and the membrane surface, enhancing the probability of E2 engaging with electroactive sites and consequently reducing the proportion of E2 mass removed by adsorption. The CNT EMR demonstrated selective degradation of micropollutants based on their molecular structures. Further research is required to enhance the degradation capabilities across a wider array of micropollutants in water treatment.

These findings highlighted the innovative nature of integrating UHPLC-FSA and LSC techniques to investigate the fundamental mechanisms of the complex EMR processes that involving an intricate interplay of multiple reactions—adsorption/desorption, degradation and byproduct transformaiton—that occur concurrently. This method can be readily applied to other EMR systems, such as those without adsorption/desorption or byproduct formation.

## Methods

This work employed a filtration cell capable of applying electric potential within realistic hydrodynamics that can reflect the complexities of fluid conditions in practical filtration applications, as well as analytical tools adept at detecting micropollutants at trace concentrations and differentiating between intact and degraded SHs.

### Electrochemical filtration system and protocol

The electrochemical filtration experiments detailed in this study were conducted using a custom-built flow-through EMR system with a commercially available membrane cell, as illustrated in Fig. 9.

Central to this system is a commercial electrochemical filtration cell (Supplementary Fig. 13., model CF016A, SterliTech, USA[71]). This cell is designed to accommodate an electrochemical membrane as a flow-through anode, and a stainless steel plate as a cathode, providing an effective filtration area of 20 cm². Additional details of the cell are provided in the Supplementary Information (SI).

Other key components of the setup include: a direct current power supply (0–30 V, 0–5 A, DSP-3005, VoltCraft, Germany) to regulate the voltage across the electrochemical cell, and a peristaltic pump (07528-30, MasterFlex, USA) equipped with a pump head (77252-72, Cole-parmer, MasterFlex, USA) and a Versilon™ A-60-N tubing (inner diameter 1/8 inch, 06504-16, MasterFlex) that allows flow rate ranging from 0 to 92 mL min⁻¹, a 16-port switching valve (Azura v.4.1, KNAUER), which facilitates the automatic collection of permeate samples, a balance (0-2200 g, AX2202, Ohaus, Germany) is employed to measure the mass of the permeate, and a data acquisition card (DAQ, model USB-6211, National Instrument, USA) is used to acquire and transfer data from the various sensors and the balance to a computer for analysis. The operation of the peristaltic pump, power

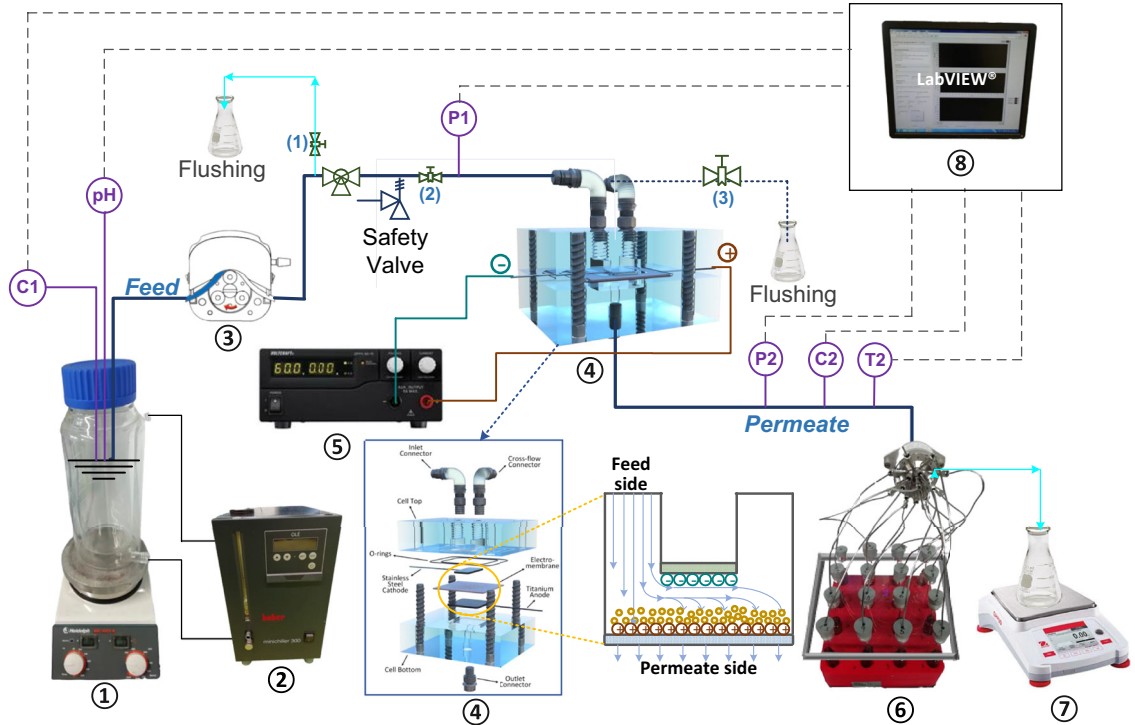

**Fig. 9 | Schematic diagram of the electrochemical filtration system.** ① Feed tank, ② ciller, ③ peristaltic pump, ④ electrochemical filtration cell, ⑤ power source, ⑥ switching valve, ⑦ balance, ⑧ data acquisition card (DAQ). C: conductivity, pH: pH, P: sensor, T: temperature sensors.

supply, and switching valve is integrated and automated through a LabView program interface (version 20.0.1, National Instruments, USA). Further details on the system are available in the SI.

The electrochemical filtration trials were executed using a single-pass dead-end filtration approach, following a detailed experimental protocol as outlined in Supplementary Table 1 to ensure accuracy and consistency. Prior to each experiment, the permeability of the membrane was assessed by measuring pure water fluxes. Subsequently, the membrane was pre-adsorbed with 500 mL of SHs feed solution without imposing a cell voltage. Afterwards, the electric power was activated, and another 500 mL of feed was subjected to treatment. Unless otherwise noted, the standard parameters for these experiments were set to a1000 mL total filtration volume, 1.6 V cell voltage, 150 L m$^{-2}$ h$^{-1}$ flux (equivalent to a flow rate of 5 mL min$^{-1}$), pH 8.2 ± 0.2, 23 ± 1 °C, and SHs initial concentration 1 μg L$^{-1}$.

### Electrochemical membrane preparation and characterization

The basic elements of the design and operations of CNTs electrochemical membranes were prepared using the method developed by Zhu et al.[72]. In short, a solution containing 0.1 g L$^{-1}$ CNTs (99 wt%, outer diameter of 13–18 nm, tube length of 3-30 μm, and –COOH content of 7%, Cheaptubes Inc., USA) and 1 g L$^{-1}$ dodecylbenzene-sulfonic acid sodium salt (DDBS, technical grade, Sigma-Aldrich, USA) was sonicated for 30 min (450 Digital Sonifier, USA), followed by centrifugation (Avanti J-E Centrifuge, Beckman Coulter, USA) at a relative centrifugal force (rcf) of 11,000x $g$ to remove undispersed particulates. After this, 75 mL of the CNT suspension were pressure-deposited on the PES membrane (Synder Filtration, USA) support with a molecular weight cut off of 20 kDa (corresponding to membrane pore diameter about 3.8 nm) using a dead-end filtration cell at 4 bar.

The permeability of the membrane was measured using MilliQ water, demonstrating to be 218 ± 1 L m$^{-2}$ h$^{-1}$.bar (Supplementary Fig. 14.). Based on scanning electron microscopy (SEM, FEI XL30 SEM-FEG, Hillsboro, USA) analysis of the membrane cross-section, the thickness of the formed CNT layer was approximately 2 μm. The pore diameter was determined to be around 0.125 μm, as inferred from the voids identified in the SEM images of the CNT layers[72–74]. This measurement aligns with prior research indicating that the pore sizes of a fiber network typically span 6 to 8 times the diameter of the fibers[73]. The electric conductivity of the CNTs membrane surface was measured to be 1389 S m$^{-1}$ with a four-point probe (4PP, 2611 A, KEITHLEY, USA, Supplementary Fig. 15) through Eq. (S2).

Surface charges (Supplementary Fig. 16) of the CNT membranes were determined in the electrolyte containing 10 mM NaCl with a SurPASS Analyzer (Surpass, Anton Paar, GmbH, Graz, Austria).

### Measurements of membrane surface potential

The design of the commercial EMR did not allow incorporating a reference electrode for in-situ control of the anodic potential applied on the CNT membrane. The membrane surface potential *vs*. an Ag/AgCl reference was determined at varying cell voltage ex-situ using the open circuit potential method[75]. The measurements were performed with the electrolyte containing 10 mM NaCl and 1 mM NaHCO$_3$, using a potentiostat (Zennium Pro, Zahner, Germany) in a three-electrode setup (Plate Material Evaluating Cell, Teflon, ALS, Japan, Supplementary Fig. 17) with the CNTs membrane as the working electrode (anode), Pt wire (Φ0.5 mm, ALS, Japan) as the counter electrode (cathode), and an Ag/AgCl electrode (Redox.me, Sweden) as the reference electrode. The surface potential of the CNT membrane *vs*. Ag/AgCl at the standard cell voltage (1.6 V) was determined to be 1.36 V in 10 mM NaCl and 1 mM NaHCO$_3$ (Supplementary Fig. 18).

### Steroid hormone and solution chemistry

Four isotope-labeled SHs: estrone (E1, [2,4,6,7-$^3$H] activity 3.689E + 12 Bq mmol$^{-1}$); estradiol (E2, [2,4,6,7-3H(N)] 3.256 TBq mmol$^{-1}$), progesterone (P, [1,2,6,7-3H(N)] 3.626 TBq mmol$^{-1}$), and testosterone (T, [1,2,6,7-3H(N)] 2.941 TBq mmol$^{-1}$) were used. Detailed characteristics of these hormones are provided in Supplementary Table 3. These

hormones were supplied by Perkin Elmer (USA) in pure ethanol (EtOH) solutions. To prepare the stock solution, the native SHs were diluted in MilliQ water to achieve a concentration of $10 \mu g \, L^{-1}$. Subsequently, feed solutions with a concentrations of $1 \mu g \, L^{-1}$ of SHs were prepared. This concentration is an order of magnitude higher than the typical SH concentration of $100 \, ng \, L^{-1}$ in aquatic environments. The preparation involved mixing radiolabelled SHs ($100 \, ng \, L^{-1}$) with non-labelled SHs, obtained from Sigma-Aldrich (USA), in a background electrolyte solution composed of $1 \, mM \, NaHCO_3$ (99.7% purity, VWR Chemicals, Germany) and $10 \, mM \, NaCl$ (99.9% purity, VWR Chemicals, Germany). Given the low solubility of SHs in water, the non-labelled SHs were prepared in $1 \, mg \, L^{-1}$ of methanol (MeOH, 99.7%, Fischer Scientific, Germany), resulting in the introduction of $79.2 \, mg \, L^{-1}$ MeOH into the feed solution. Presence of $HCO_3$, EtOH and MeOH in the electrolyte can act as scavenger towards ·OH, which must be noted in these experiments.

Feed solutions exceeding $1 \mu g \, L^{-1}$ of SHs, a combination of radiolabelled SHs ($100 \, ng \, L^{-1}$) and proportionate volumes of non-labelled SHs, prepared in pure MeOH across concentrations ranging from 1 to $10,000 \, mg \, L^{-1}$, was employed. This approach ensures a consistent concentration of MeOH at $79.2 \, mg \, L^{-1}$ and EtOH at $27.2 \, mg \, L^{-1}$ in the feed solutions, irrespective of SH concentration variations. This strategy was adopted to maintain a consistent scavenging effect throughout all experiments, which cannot be eliminated.

### Analytical methods

A methodological approach, integrating a liquid scintillation counter (LSC, 2550 TR/AB, Packard, USA) and an ultra high-performance liquid chromatograph (UHPLC, Flexar, Perkin Elmer, USA) coupled with a flow scintillator analyzer (FSA, Radiomatic 625TR, Perkin Elmer, USA), was employed to elucidate the mechanisms of the SH removal by EMR, with a particular focus on the concurrent behavior of degradation and adsorption.

The UHPLC-FSA is utilized to achieve high-resolution separation of the sample constituents, facilitating the identification and quantification of both the intact SH compound and its degradation products potentially forming in the system. This separation is crucial for understanding the chemical dynamics of the SH during electrochemical reactions. Concurrently, LSC quantified the total $^3H$ activity of the samples, offering a comprehensive measurement of the entire radiolabelled species present, irrespective of their state as either the parent compound or degradation products. This synergistic integration of techniques affords a detailed understanding of the complex chemical and physical transformations within the electrochemical matrix.

The UHPLC-FSA analytical method, including elution parameters detailed in Supplementary Table 4, was adapted from Lyubimenko et al.[76]. Modifications included adjusting the elution mobile phase flow rate from 0.25 to $0.2 \, mL \, min^{-1}$, and increasing the injection volume from 100 to $200 \mu L$. These adjustments allowed the method to operate at lower pressures without compromising analytical quality. The calibration curve (Supplementary Fig. 19. A) demonstrated a limit of detection (LOD) of $2.5 \, ng \, L^{-1}$ for UHPLC-FSA, which is comparable to the LOD of $1.5$-$2.4 \, ng \, L^{-1}$ reported by Lyubimenko et al.[76].

For LSC analysis, samples were prepared by mixing 1 mL of aqueous sample with 1 mL of Ultima Gold LLT scintillation cocktail (Perkin Elmer, USA), and $^3H$ activity measurements were conducted over 10 minutes in triplicate[77]. The calibration curve for LSC (Supplementary Fig. 19. B), demonstrated a strong linear correlation for E2 across a concentration range of 0.1 to $100 \, ng \, L^{-1}$, which indicated a LOD below $0.1 \, ng \, L^{-1}$.

### Data processing and error propagation

Calculation of the surface resistance (S $m^{-1}$) of the CNT membrane, pure water flux ($J_w$, L $m^{-2}$ $h^{-1}$), permeability ($L$, L $m^{-2}$ $h^{-1}$ $bar^{-1}$), and mean hydraulic residence time ($t_r$, s) of the CNT layer on the electrochemical membrane are given in Supplementary Table 5.

SH removal (R, %) was calculated using Eq.(1), which evaluates the removal efficiency of the EMR by the combined electrochemical adsorption and degradation at steady-state conditions.

$$R = \left(1 - \frac{c_{p,eq}}{c_f}\right) \cdot 100 \qquad (1)$$

where $c_f$ (ng $L^{-1}$) is the initial concentration of SH in the feed, $c_{p,eq}$ (ng $L^{-1}$) is the SH concentration in the permeate at equilibrium state, determined at the end of the experiment when the cumulative permeate volume ($V_p$) reaches 1000 mL. To address potential inaccuracies caused by data point fluctuations in calculating $R$, $c_{p,eq}/c_f$ value was defined by fitting the experimental data of $c_{p,eq}/c_f$ $vs.$ $V_p$ to a power model ($y = a \cdot x^b$). The specifics of this fitting procedure are adapted from previous literature[55] and detailed in Supplementary Fig. 20.

To compare the formation of degraded products, their permeate concentrations were normalized by the initial SH concentration in the feed ($c_{p,prod}/c_{f,SH}$). As the identification of the byproducts has not been possible, accurately determining the concentrations using calibration was not feasible. Therefore, the concentration of the byproducts was estimated using the calibration for the parent SH. It is important to note that these concentrations, as defined in this study, do not represent the actual concentrations of each byproduct but are used for quantitative comparisons of the formation of the same byproduct at varying conditions.

The evaluation of SH removal kinetics within the EMR relied on calculating the apparent rate of SH removal ($r'_{rem}$, mol $m^{-2} s^{-1}$) reflects the total quantity of SH removed through electrochemical adsorption and/or degradation, normalized per unit area of the membrane and per unit time, across the 1000 mL filtration experiment.

$$r'_{rem} = \frac{m_f - m_p}{t \cdot M \cdot A} = \frac{m_{rem} \cdot J_w}{3600 \cdot M \cdot V} \qquad (2)$$

where $m_f$ (g) is the mass of E2 in the feed solution, $m_p$ (g) is the accumulative mass of E2 in permeate, $t$ (s) represents the time span of the filtration experiments, $M$ (g $mol^{-1}$) is the molecular weight of the SH type, $A$ ($2 \cdot 10^{-3} \, m^2$) is the effective membrane surface area, $J_w$ (L $m^{-2}$ $h^{-1}$) is the flux, and $V$ (1 L) is the total volume filtered. $m_{rem}$ (g) quantifies the total mass of SH removed over the 1 L of the filtration experiment, encompassing both adsorption and degradation phases. The calculation of $m_{rem}$ involves the curve fitting and integration of the experimental data of SH concentration in permeate ($c_p$, ng $L^{-1}$) over the permeate volume. The detailed methodology for this process is thoroughly explained in Supplementary Fig. 21.

Throughout the electrochemical filtration experiments, byproducts were produced as a result of SH degradation, and their formation rate serve as an indicator of the dynamics of chemical transformation. To assess the kinetics of SH transformation during the electrochemical degradation process, the apparent rate of byproduct formation ($r'_{prod}$, mol $m^{-2}$ $s^{-1}$) was determined using Eq. (3).

$$r'_{prod} = \frac{m_{p,3H} - m_p}{t_{deg} \cdot M \cdot A} = \frac{m_{prod} \cdot J_w}{3600 \cdot M \cdot V_{deg}} \qquad (3)$$

where $m_p$ (g) represents the total mass of tritium-labelled compounds in the permeate, including both intact SH molecules and degradation products desorbed from the membrane, $t_{deg}$ (s) is the time duration represents the period of the electrochemical degradation phase, which occurs over a cumulative permeate volume of 500 mL ($V_{deg}$) after the voltage application. The difference between $m_{p,3H}$ and $m_{p,SH}$ ($m_{prod}$) quantifies the cumulative mass of total byproducts ($m_{prod}$, g) present in the permeate during the electrochemical degradation process. Supplementary Fig. 22. illustrates the process of determining

$m_{prod}$ by fitting and integrating the data points from both UHPLC-FSA and LSC across the permeate volume. Since the permeate sample during the degradation phase comprises a mixture of SH and its byproducts, individual calibration of this sample was not feasible. Consequently, the calibration curve established for SH was employed to approximate the quantification of the total mass.

The contribution of electrochemical adsorption ($\theta_{ads}$, %) and degradation ($\theta_{deg}$, %) within the EMR to the total mass of SH in feed can be estimated via. Equations (4) and (5).

$$\theta_{ads} = \frac{m_{ads}}{m_f} \tag{4}$$

$$\theta_{deg} = \frac{m_{deg}}{m_{rem}} = \frac{m_{rem} - m_{ads}}{m_f} \tag{5}$$

where $m_{ads}$ (g) is the total mass of the SH adsorbed on the membrane over the experiment, which is derived from the data of total radio-labelled compounds measured by LSC (Supplementary Fig. 23). Subsequently, the total mass of the SH removed by degradation ($m_{deg}$, g) can be determined by the difference between the mass of total removed SH and adsorbed SH.

The estimation of errors in SH concentration was conducted using the error propagation approach[78,79], accounting for variations arising from multiple sources such as solution preparation, the filtration system, the operators, and analytical devices. The specific error sources and their respective estimated relative errors are detailed in Supplementary Table 6.

## Data availability

The data generated in this study are included in the main text and Supplementary Information files. Source data for figures are provided with this paper.

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

## Acknowledgements
The authors express their gratitude to the Helmholtz Recruitment Initiative (MembraneTechnology, A.I.S) for funding of IAMT lab and the Bundesministerium für Bildung und Forschung (BMBF) project NEMWARE (02WIL1555, A.I.S) for financial support of this project. Special thanks are extended to Mr. Mehran Aliaskari (KIT-IAMT) for his assistance in constructing the electrochemical filtration system; Dr. Mina Ahsani (KIT-IAMT) for her diligent filtration experiments concerning varying flux (Fig. 6), SH concentration (Fig. 7), and SH types (Fig. 8); Dr. Xiaobo Zhu for fabricating CNT electrochemical membranes; Dr. Akhil Gopalakrishnan (KIT-IAMT) for assistance in the measurement of zeta-potential; Prof. Bryce S. Richards and Dr. Dmitry Busko (KIT-IMT) for providing access to and guidance with the four-point probe meter essential for surface resistance measurements.

## Author contributions
A.I.S. conceived the project and provided expertise in membranes. S.L. and A.I.S. developed the concept of this work, particularly the electrochemical membrane process and analytical methods. S.L. designed and constructed the electrochemical filtration system, validated the method and developed the filtration protocol, conducted feasibility test and experiments at varying cell voltages, and performed the membrane characteraization. D.J. fabricated the electrochemical membrane and provided expertise on electrochemical membrane reactors. D.M. provided expertise on electrochemical mechanisms. S.L. wrote the manuscript. All authors have revised the manuscript.

## Funding

## Competing interests
The authors declare no competing interests.
