## [Peer Review File · Nature Communications]

Differentiation of adsorption and degradation in steroid hormone micropollutants removal using electrochemical carbon nanotube membraneREVIEWER COMMENTS

Reviewer #1 (Remarks to the Author):

This manuscript developed an analytical method to differentiate the concurrent adsorption and degradation processes in the removal of steroid hormones on a EMR by integration of high-performance liquid chromatography-flow scintillator analyzer and liquid scintillation counting techniques allows, which is important for clarifying the underlying mechanisms of electrochemical micropollutant removal. However, I think that this work focuses on a specialized field, and the model selected is not representative (see remark 1), which is not suitable for publication in a comprehensive journal like Nature Communications in its current form. It is more suitable for publication in a specialized journal, such as Analytical Chemistry. Here are some technical issues for improving the quality of this work.

1. The authors selected a CNT-based membrane as a model EMR, which has a high surface area and strong lipophilicity. Therefore, it usually has strong affinity for organic micropollutants and thus high adsorption capability. However, this model may not be applicable to other EMRs, such as metal and/or metal oxide membranes. In addition, the authors found that different micropollutants have different adsorption and degradation processes even on the same EMR, letting alone the comparison of various supports. This work does not provide a viable solution to clarify the intricate interplay among adsorption, desorption and degradation processes.
2. High-resolution LC-MS should be employed to determine the exact structure of the by-products (e.g., 3m, 5m and 8m), which will be beneficial for understanding the underlying degradation mechanism.
3. The introduction and conclusion should be more concise, highlighting the core arguments.
4. The applied cell voltage may influence not only the electron transfer rate but also the degradation mechanism, e.g., direct oxidation or indirect reactive species, resulting in different adsorption and degradation processes. The byproduct-5m may result from indirect degradation processes rather than from the faster electron transfer rate. How do the reactive species affect the adsorption processes? To clarify these processes, quenching experiments for active species should be performed.

5. "Conversely, about $24\pm 10\%$ of the E2 remained in the permeate. This residual E2, constituting $19\pm 4\%$ of the total, was ascribed to the fraction that did not undergo adsorption during the adsorption phase." How did the authors calculate ($19\pm 4\%$) and reach this conclusion? Please detail it.
6. "Given that all E2 molecules were adsorbed onto the membrane (with E2 concentration in the permeate being < 2.5 ng/L) during the degradation phase,....." This assumption may also be untenable because indirect active specie oxidation does not require the reactants to be adsorbed on the electrode surface.
7. The horizontal coordinates of Figure 2a,b and Figure 8a,b are too close. The permeate volume of 1000 in Figure a is easily misread as 10000.
8. Some characterizations related to used CNTs and membrane properties such as morphology, pore distribution, and pure water flux, etc. should be added.

Reviewer #2 (Remarks to the Author):

The manuscript by Liu et al. describes a method to differentiate adsorption and degradation of contaminants on an electrochemical membrane. By using HPLC-FSA and LSC together, the authors measured the concentrations of the parent compounds and the transformation products. The measurements of both parent compounds and transformation products can be used to conduct a mass balance analysis, which differentiates the adsorption and degradation. With this method, Liu et al. assessed the performance of a CNT-membrane under different operating conditions, including applied potentials, water fluxes, and contaminant concentrations.

The experiments are comprehensive, and the data analysis is sound. The authors provided detailed analysis on error estimation regarding their mass balance.

I recommend acceptance of the manuscript after the authors address the following comments.

Major comments:

- 1) The authors consider adsorption/desorption and reactions as potential rate-limiting steps. However, for electrochemical reactive membranes, the rate-limiting step could be the transport of contaminants to the active sites, which motivates the development of flow-through electrodes. Please comment on the importance of adsorption/desorption versus mass transport of contaminants.
- 2) Based on the method proposed by the authors, the fraction of contaminants being removed versus being adsorbed should be a function of time (as shown by the authors in Figure 2). Over extended continuous operation, the contribution from sorption should be minimal in comparison to degradation. The authors should provide information on how time is being considered in their analysis and how the results are meaningful in terms of long-term and realistic operation.
- 3) The authors performed their experiments using CNT electrode which has a high adsorption capacity for contaminants. To make the conclusions more universal, the authors are recommended to discuss further about other electrode materials, which may have lower adsorption capacity than CNT.
- 4) Line 150: Please provide more explanations on why E2 can possibly get electrochemically desorbed after applying potentials. The E2 should not be charged under neutral pH, assuming NaHCO₃ was used as a buffer to maintain circumneutral pH conditions.
- 5) Figure 3B: It's unclear how the authors normalized the permeate concentrations of the transformation products by the feed concentrations of the transformation products. The feed concentrations of the transformation products should be zero.
- 6) Line 158-162, the authors claimed this is an effective technology for treatment. However, high concentrations of oxidation products are released as evidenced in Figure 3 and 4. The authors should comment on the inability of this system in mineralizing contaminants.
- 7) Figure 4C: For the analysis with pre-adsorption, the authors should provide explanations on the rationale of taking the two "in permeate (unremoved)" as two independent pathways. They are basically the residual after adsorption and reaction, why this unremoved part under no potential and the one with potential can be taken as individual process and can be stacked together for mass balance? Is this because the integration of the area under the curve for these two masses are conducted in separate time periods? If so, please clarify the manuscript.
- 8) Figure 4C, why is the in permeate (power off) and adsorbed on membrane prior to

applying voltage adds up to 50%? Also, can the authors please check if the bottom right label “in permeate (unremoved)” is labelled correctly with the arrow pointing to the blue bar?

9) Line 1107: The authors should justify their assumptions that all byproducts are fully desorbed. This assumption is crucial for the analysis of mass balance. However, there is no experimental evidence (e.g., extraction) to confirm this assumption.

10) Figure 7, can the authors provide explanations on why there was no sorption for the 105 ng/L experiment?

Minor comments

(1) Line 99, the sentence is not complete. “Formation of active (missing word) in electrochemical processes”

We greatly appreciate the reviewers' thoughtful comments and your consideration of our manuscript following its revision. The reviewers' comments have been addressed, specific responses to reviewer comments are provided below, and the corresponding changes made in the manuscript are highlighted in this revision. Some minor revisions have also been made in the manuscript to improve the language, these changes are also highlighted.

COMMENTS FROM THE REVIEWERS

Reviewer#1:

This manuscript developed an analytical method to differentiate the concurrent adsorption and degradation processes in the removal of steroid hormones on a EMR by integration of high-performance liquid chromatography-flow scintillator analyzer and liquid scintillation counting techniques allows, which is important for clarifying the underlying mechanisms of electrochemical micropollutant removal. However, I think that this work focuses on a specialized field, and the model selected is not representative (see remark 1), which is not suitable for publication in a comprehensive journal like Nature Communications in its current form. It is more suitable for publication in a specialized journal, such as Analytical Chemistry. Here are some technical issues for improving the quality of this work.

Thank the reviewer for the comments. However, we respectfully disagree with the assertion that the model selected (we assume the reviewer refers to the chosen micropollutant group of steroid hormones and the CNT materials?) is unrepresentative and unsuitable for publication in a comprehensive journal like Nature Communications. Analytical Chemistry is far from our field of water process engineering and does not suit our work at all! The Schäfer team has worked with steroid hormones in environmentally relevant concentrations (usually 100 ng/L) since 2000 and developed extensive expertise in analyzing small volumes of very low concentrations at high accuracy. This experience has enabled this work and will have to be established prior to working with other micropollutants. The tools are then applicable to many materials. While many material groups shortcut this difficult requirement by working at higher concentrations, this creates a bias in results and omits exactly what the reviewer is concerned about – adsorption effects – that are only measurable for micropollutants in very low concentrations, where mass transfer dominates behaviour. A detailed explanation has been provided in our response to Comment 1.

Comment 1: The authors selected a CNT-based membrane as a model EMR, which has a high surface area and strong lipophilicity. Therefore, it usually has strong affinity for organic micropollutants and thus high adsorption capability. However, this model may not be applicable to other EMRs, such as metal and/or metal oxide membranes. In addition, the authors found that different micropollutants have different adsorption and degradation processes even on the same EMR, letting alone the comparison of various supports. This work does not provide a viable solution to clarify the intricate interplay among adsorption, desorption and degradation processes.

We appreciate the reviewer's comment. We agree that the examined EMR model may not be applicable to other electrode materials, such as metal and/or metal oxide. Naturally, material properties play an important role. However, the primary objective of this study was to address the significant challenge of micropollutant removal from water, specifically focusing on endocrine-disrupting steroid hormones (SHs), due to their pervasive presence and the substantial difficulties associated with their removal. These micropollutants are biologically active at very low concentrations, typically in the ng/L range. For example, concentrations of 17 β -estradiol (E2) as low as 4.7 ng/L have been shown to induce vitellogenin in juvenile female rainbow trout [1],

which highlights the critical need for effective removal technologies, reflected in the proposed guideline value of 1 ng/L for E2 in drinking water [2]. EMR, as an advanced emerging technology [3], shows considerable promise in addressing this issue, also because of the high affinity of these CNT for SHs. To the best of our knowledge, there have so far been no studies that investigate the performance of membrane electrocatalysis for a meaningful concentration range of micropollutants – that is for ng/L or lower. Only a few studies [4, 5] have so far reported the degradation of micropollutants by EMR at an unrealistic concentration (> 100 µg/L), which can create a misleading perception of pollutant risk mitigation. For instance, studies conducted by dos Santos Cunha *et al.* [4] employed CNT membranes for the degradation of E2 and 17α-ethinylestradiol (EE2) at initial concentrations of 10 and 11 mg/L respectively, achieving 99% removal but resulting in effluent concentrations around 100 µg/L, where the initial concentration is above the solubility in water! Such concentrations significantly exceed the guideline values and pose serious risks to human and aquatic health. Furthermore, it has been reported that these high initial concentrations can lead to an overestimation of the performance of electrochemical membranes due to increased diffusion rates driven by larger concentration gradients [6]. This overestimation hampers the accurate application of these results to the remediation of most real-world water sources such as groundwater, surface water, and wastewater effluents. Given these challenges, we developed and employed an innovative analytical method, UHPLC-FSA coupled with LSC, capable of detecting SH micropollutants and their degradation products at ng/L levels. This methodology enables us to investigate the behavior of SH micropollutant removal within the EMR under realistic, environmentally relevant conditions and even differentiate between degradation and adsorption. All scientists who work in this field know how excruciatingly difficult this is, and we think this is completely necessary and suitable for a comprehensive journal like Nature Communications.

On the other hand, CNTs with their high surface area and adsorption capacity have been documented to be an excellent material for the electrochemical treatment of organic micropollutants. Accordingly, this work focused on the removal of endocrine active neutral micropollutants using a CNT membrane. As clearly stated in our manuscript, the scope of this work was specifically centered on the adsorption and degradation of SH on CNTs, which per se represent a complex and important group of novel materials, rather than on the development of a universal model applicable to all materials – which would be impractical given the intrinsic diversity in electronic structures, surface functionalities, and other properties of various material groups or their combinations.

We also respectfully disagree with the reviewer's assertion that this work does not provide a viable solution to clarify the intricate interplay among adsorption, desorption and degradation processes. For instance, most previous studies of the EMR process required pre-saturation of the membrane with micropollutants to ascertain the role of degradation in total removal. The degradation kinetics of pre-adsorbed micropollutants are likely different, and pre-saturation complexes the understanding of the degradation dynamics because of the interplay of multiple processes. In this study, the adsorption and degradation processes were distinguished through mass balance analysis in section '2.3. Role of pre-adsorption'. This analysis revealed that the pre-adsorption was not necessary for improving removal by electrochemical degradation—likely due to the high affinity of SH with the CNT surface and the enhanced mass transport in the EMR—contrary to the common belief that it enhances electrocatalytic degradation. While pre-adsorption may play a critical role in electrocatalysis by other materials with lower surface area and affinity, the method developed in this study can also identify its effects. Thus, we believe that the methods and results presented are crucial for the design and development of the EMR process for the efficient removal of micropollutants in practical applications. Furthermore, the investigation of '*Electrochemical degradation of different types of steroid hormones*' (Section 2.7), provides clear evidence of the dynamic interplay between adsorption, desorption, and

degradation processes for different hormone types. For example, E2 exhibited high adsorption on the CNT membrane, followed by rapid degradation. While E1 showed adsorption levels as high as E2, it was subsequently removed mainly by electro-adsorption rather than degradation. Importantly, adsorption was identified as a prerequisite step for the electrochemical process to commence. Meanwhile, the adsorbed testosterone (T) and progesterone (P) could be degraded due to the nature of their chemical structures. These processes occur simultaneously during filtration and electrocatalysis, making the experimental methods needed to ascertain their contributions extremely challenging. The success achieved in this study in quantifying concurrent adsorption and degradation is very novel.

Comment 2: High-resolution LC-MS should be employed to determine the exact structure of the by-products (e.g., 3m, 5m and 8m), which will be beneficial for understanding the underlying degradation mechanism.

We thank the reviewer for the valuable suggestions regarding the byproducts analysis by LC-MS. High resolution LC-MS should indeed be employed to determine the exact structure of the degraded products. However, the development of the high-resolution LC-MS method for analyzing these specific by-products at extremely low concentrations after treatment poses critical challenges. Though LC-MS is a highly sensitive analytical technique, the detection of limit (LOD) can still be a limiting factor when analyzing trace amounts of complex byproducts. Prior studies have reported the LOD of 17-7317 ng/L for various SHs [7, 8]. The very low concentrations and complex matrix after electrochemical treatment make the detection of byproducts formed as a result of their degradation, especially the byproducts-3m and -5m, very difficult. The concentration of relevant samples could and does induce significant errors, as we learned from our experience. Additionally, for accurate identification and quantification using LC-MS, standards of the byproducts are typically required. In many cases, especially in the case of the formation of unidentified degradation byproducts, these standards may not be readily available. The establishment and validation of such a method require extreme time and resources, to achieve the necessary reliability and reproducibility of the results, adding layers of complexity and effort to the analytical process.

Given these complexities, and considering the primary objectives of this study, we have referred in this aspect to the literature that proposed the possible degraded products via electrochemical oxidation (Figure R1).

Figure R1. Pathway and byproducts of electrochemical degradation of estradiol via direct electron transfer, reported in the literature [9-11].

In the manuscript, the discussion regarding the byproducts was described in Lines 198-199, '*Prior studies 68-70 have identified a ketone derivative formed as a product of the direct electrochemical oxidation of E2 through a two-electron transfer mechanism, as evidenced by GC-MS analysis It is posited that the ketone derivative corresponds to the peak observed at 8 minutes in the UHPLC-FSA chromatograms. This derivative subsequently undergoes further oxidation, yielding a range of cleaved byproducts that were eluted between 2 to 4 minutes in the UHPLC analysis*'.

Comment 3: The introduction and conclusion should be more concise, highlighting the core arguments.

We thank the reviewer for the comment. As the reviewer suggested, the Introduction and Conclusion have been modified, which has been highlighted in the manuscript.

In the Introduction, the discussion explaining the risks associated with SH micropollutants, the fundamentals of electrochemical oxidation technology, and the mechanisms related to the in-situ generation of secondary reactive species (which are not relevant in this work) has been succinct.

Comment 4: The applied cell voltage may influence not only the electron transfer rate but also the degradation mechanism, e.g., direct oxidation or indirect reactive species, resulting in different adsorption and degradation processes. The byproduct-5m may result from indirect degradation processes rather than from the faster electron transfer rate. How do the reactive species affect the adsorption processes? To clarify these processes, quenching experiments for active species should be performed.

We thank the reviewer for the thoughtful comment. We agree with the reviewer that the indirect oxidation could be an alternative mechanism for SH degradation. The oxidation of water on the CNT surface could lead to the in-situ generation of hydroxyl radical ($\cdot\text{OH}$), which is a powerful radical in the oxidative processes [12]. Additionally, active chlorine, a distinct category of effective oxidizing agents including Cl_2 , HClO , and ClO^- , $\text{Cl}\cdot$, and $\cdot\text{Cl}_2$, is derived from the electrochemical oxidation of Cl^- present in the solution to generate [13, 14], as described *via* reactions Eq. (1)-(4) [15].

Indeed, we have identified and quantified the $\cdot\text{OH}$ and active chlorine produced on the CNT surface using various methods.

The production of active chlorine was assessed using the iodometric method [16], which involves the oxidation of iodide ions (I^-) into iodine (I_2). Briefly, a 20 mM potassium iodide (KI) solution was prepared in a buffer solution comprising 0.03 mM of sodium acetate anhydrous (>99% purity, Merck, Germany) and 0.374 mM of acetic acid (100% purity, Merck, Germany), yielding a KI stock solution with an excess of I^- ions. Subsequently, 5 mL of the sample was added into 5 mL of the KI stock solution, where active chlorine underwent a reaction with the I^- to produce I_2 (Eq. (5)). The resulting I_2 further reacts with the surplus I^- , forming tri-iodide (I_3^-) ions (Eq. (6)). Quantification of the resultant I_3^- was performed via measurement of absorbance at 352 nm using a UV-vis spectrometer (Lambda 25, Perkin Elmer, USA). The mixed solutions were left to stand for 5 min before measuring the absorbance.

Prior to the measurement, this method was validated using active chlorine electrochemically generated in a batch system consisted of two platinum wire electrodes (50HX15 0.6/250MM, Redox.me, Sweden) with a distance of 0.5 cm in 15 mL of 10 mM NaCl electrolyte. Varying concentrations of active chlorine solution were

achieved by applying varying cell voltage for 60 s. Figure R2 demonstrates that the absorbance of the sample at 352 nm increased consistently as the voltage increased from 1 to 20 V, indicating that the iodometric method is applicable for quantitative measurement of active chlorine.

Figure R2. Absorbance spectra for a mixture of KI/NaAc/HAc solution and active chlorine solution electrochemically generated at varying cell voltage from 1 to 20 V.

Figure R3 shows the absorbance of the sample collected from the CNT EMR at varying cell voltage, where negligible changes in the absorbance are observed in the examined range. This implies that no or minimal amounts active chlorine are produced by the CNT membrane even at a voltage of up to 20 V.

Figure R3. Absorbance spectra of permeate produced using the filtration of KI/NaAc/HAc solution at varying cell voltage from 0.9 to 3 V within the CNT EMR. $J_f = 600 \text{ L/m}^2\text{h}$ (2 mL/min), 10 mM NaCl, 1 mM NaHCO₃, 27.2 mg/L EtOH, 79.2 mg/L MeOH, pH 8.3 ± 0.3 , $23 \pm 0.2 \text{ }^\circ\text{C}$.

The production of $\cdot\text{OH}$ was quantitatively measured at varying cell voltages using a reported coumarin-mediated probing method. Coumarin has been widely used for detecting $\cdot\text{OH}$, facilitating the generation of a fluorescent product, 7-hydroxycoumarin (7-OHCoumarin), for $\cdot\text{OH}$ quantification (Eq.(7)) [17]. To perform the measurement, a feed solution containing 0.01 mM coumarin was subjected to filtration using the CNT membrane with the voltage on. The byproduct, 7-hydroxycoumarin, resulting from the hydroxylation reaction between coumarin and $\cdot\text{OH}$ was then measured by a fluorescence spectrophotometer (Cary, Eclipse Varian,

Agilent, US) at excitation wavelength (λ_{ex}) of 330 nm [18], allowing for a detection limit (LOD) of 5 nM for the $\cdot\text{OH}$ [19].

To validate this method, the $\cdot\text{OH}$ production with a TiO_2 -coated polyethersulphone (PES- TiO_2) photocatalytic membrane [20], that are known for effectively generating $\cdot\text{OH}$, was measured. Figure R4 showed a peak at around 375 nm for all fluorescence spectra, which was ascribed to the water scattering. The small feature in the range between 380 and 450 was ascribed to the parent coumarin. A peak assigned to $\cdot\text{OH}$ -coumarin at 450 nm was observed, indicating the feasibility of using coumarin as a probe for the detection of $\cdot\text{OH}$.

Figure R4. Changes of fluorescence spectra in permeate with cumulative permeate volume (V_p) over the photocatalytic filtration. PES- TiO_2 , $c_f(\text{coumarin}) = 0.01 \text{ mM}$, $I_{inc} = 10 \text{ mW/cm}^2$, 365 nm , $J_f = 600 \text{ L/m}^2\text{h}$ (2 mL/min), pH 8.3 ± 0.3 , $23 \pm 0.2 \text{ }^\circ\text{C}$.

The production of $\cdot\text{OH}$ with the CNT membrane was measured at cell voltage varying from 0 to 3 V. Figure R5 showed that within the explored voltage range, a negligible amount of $\cdot\text{OH}$ formation was detected on the CNT membrane, which agrees well with previous studies [3]. The sp^2 carbon-based electrodes are typical examples of active anodes and exhibit an overpotential for O_2 evolution generally lower than 0.4 V [21]. The widely accepted model of anodic oxidation proposed by Comninellis [22] and slightly modified afterwards by Marselli *et al.* [23], suggested that the so-called active anodes (M), with a low O_2 -overpotentials, interact strongly with electrogenerated $\cdot\text{OH}$ to form a higher state oxide (MO), $M(\cdot\text{OH}) \rightarrow \text{MO} + \text{H}^+ + e^-$, that in combination with the anode surface M (redox couple MO/M) acts as selective mediator in degrading organic compounds via direct oxidation.

Figure R5. Fluorescence spectra of $\cdot\text{OH}$ adduct in permeate for electrochemical filtration with 200 mL of coumarin solution at varying cell voltage from 0.9 to 3 V within the CNT EMR. $c_f(\text{coumarin}) = 0.01 \text{ mM}$, $J_f = 600 \text{ L/m}^2\text{h}$ (2 mL/min), 10 mM NaCl, 1 mM NaHCO_3 , 27.2 mg/L EtOH, 79.2 mg/L MeOH, pH 8.3 ± 0.3 , 23 ± 0.2 °C.

These results suggested that active chlorine and $\cdot\text{OH}$ contribute minimally to the degradation of E2, and direct electron transfer should be the predominant mechanism. To verify this, NaNO_3 was employed as a scavenger to target surface electrons (Figure R6). The result showed that negligible E2 removal was observed in the presence of 10 mM NaNO_3 , with no production of the byproducts during degradation.

Figure R6. Electrochemical degradation of E2 in the presence of scavenger NaNO_3 for surface electrons within the CNT EMR, as normalized concentration of (A) E2 and ^3H , and (B) byproducts, vs. accumulated permeate volume. $c_{f,E2} = 100 \text{ ng/L}$, $c_{f,\text{NaNO}_3} = 10 \text{ mM}$, $V_{\text{cell}} = 1.6 \text{ V}$, $J_f = 600 \text{ L/m}^2\text{h}$ (2 mL/min), 1 mM NaHCO_3 , 10 mM NaCl, 27.2 mg/L EtOH, 79.2 mg/L MeOH, pH 8.2 ± 0.2 , 23 ± 0.2 °C.

Overall, these results suggest that direct electron transfer dominates the degradation mechanism of E2 in the CNT membrane, but investigation of the reaction mechanisms is certainly not straightforward and in our opinion not within the scope of this paper. Consequently, this will be published in a separate paper to be submitted shortly. To clear the concerns involving the possible indirect mechanisms with the CNT membrane, an additional description has been added in the manuscript:

Introduction, Line 102-106, *'CNTs, which typically exhibit a low overpotential for O₂ evolution (generally < 0.4 V⁶³), are considered as active anodes (M). These materials interact strongly with electrogenerated ·OH to form a higher-state oxide (MO), M(·OH) → MO + H⁺ + e⁻, that in combination with the anode surface acts as a selective mediator in degrading organic compounds via direct oxidation^{64, 65}'.*

We hope that these explanations satisfy the reviewer.

Comment 5: "Conversely, about 24±10% of the E2 remained in the permeate. This residual E2, constituting 19±4% of the total, was ascribed to the fraction that did not undergo adsorption during the adsorption phase." How did the authors calculate (19±4%) and reach this conclusion? Please detail it.

The total mass of E2 in the feed solution (1 L) was 1030 ng (1030 ng/L x 1 L). 500 mL of the feed was filtered with the voltage off (adsorption phase), and another 500 mL was subsequently filtered with the voltage on (degradation phase), as was described in the experimental protocol (Table S1) in the manuscript. During the adsorption phase, 191 ng E2 passed through the membrane and was released into the permeate, as determined by integrating the experimental data of c_p over the permeate volume in the range of 0-500 mL (described in SI, Section 5.3), which accounts for about 19% of the initial E2 mass (191 ng/1030 ng). Using the same method, the E2 mass released into the permeate during the degradation phase (500-1000 mL) was quantified to be around 5% of the initial mass. Thus, a totally 24% of E2 remained in the permeate after the treatment, wherein 19% was released during the adsorption phase.

To clarify this, an explanation has been added to the manuscript:

Line 161, *'Details of the mass balance analysis can be found in SI (Figure SA)'.*

Supporting Information, *'5.6 Mass balance analysis for the pre-adsorption experiments' (see the response to comment 7 for reviewer#2).*

Comment 6: "Given that all E2 molecules were adsorbed onto the membrane (with E2 concentration in the permeate being < 2.5 ng/L) during the degradation phase,....." This assumption may also be untenable because indirect active specie oxidation does not require the reactants to be adsorbed on the electrode surface.

Please see the response to comment 4. In our experience, adsorption to the surface or having micropollutants near the surface where ROS are generated for indirect oxidation is indeed very important as for micropollutants the reaction is mass transfer limited.

Comment 7: The horizontal coordinates of Figure 2a,b and Figure 8a,b are too close. The permeate volume of 1000 in Figure a is easily misread as 10000.

Thank you for the comment. The x-axis scale of Figures 2 and 8 has been revised, as below.

Figure 2. Adsorption and electrochemical degradation of E2 expressed as normalized (A) total tritium (^3H) activity ($C_{p,3H}/C_{f,3H}$) measured via LSC, and (B) concentration of E2 ($C_{p,E2}/C_{f,E2}$) measured via UHPLC-FSA in permeate vs. accumulated permeate volume. $C_{f,E2} = 1 \mu\text{g/L}$, $V_{\text{cell}} = 1.6 \text{ V}$, $J_f = 150 \text{ L/m}^2\text{h}$ (5 mL/min), 1 mM NaHCO_3 , 10 mM NaCl , 27.2 mg/L EtOH , 79.2 mg/L MeOH , pH 8.2 ± 0.2 , $23 \pm 1 \text{ }^\circ\text{C}$.

Figure 8. Electrochemical degradation of individual SHs, expressed as normalized concentration of (A) SH, and (B) ^3H vs. accumulated permeate volume. $c_{f,SH} = 1 \mu\text{g/L}$, $V_{cell} = 1.6 \text{ V}$, $J_f = 150 \text{ L/m}^2\text{h}$ (5 mL/min), 1 mM NaHCO_3 , 10 mM NaCl , 27.2 mg/L EtOH , 79.2 mg/L MeOH , pH 8.2 ± 0.2 , $23 \pm 1 \text{ }^\circ\text{C}$.

Comment 8: Some characterizations related to used CNTs and membrane properties such as morphology, pore distribution, and pure water flux, etc. should be added.

The morphology of the CNT membrane was comprehensively characterized and published in previous work [24-26] (Figure R7). Thus, we referred to this studies in the Method Section, Line 491-494, 'Based on scanning electron microscopy (SEM, FEI XL30 SEM-FEG, Hillsboro, USA) analysis of the membrane cross-section, the thickness of the formed CNT layer was approximately 2 μm . The pore diameter was determined to be around 0.125 μm , as inferred from the voids identified in the SEM images of the CNT layers⁸⁴⁻⁸⁷'.

Since it has not been possible to prepare an independent CNT layer without a substrate, analyzing the pore distribution of the CNT layer using methods such as Capillary Flow Porometry or Mercury Intrusion Porosimetry is unfeasible. Consequently, the pore diameter was estimated through direct observation of SEM images of the membrane's surface and cross-section. The pore size was quantitatively analyzed using image analysis software.

Figure R7. Morphology of surface and cross-section of CNT membrane. (A) Cross-section of 2 μm -thick membrane, (B) Cross-section of a 6 μm -thick membrane, (C) the surface morphology at a magnification of 25,000 X clearly showing individual, non-aggregated CNTs, and (D) a Photo of a CNT membrane surface. Adapted from [24].

As the reviewer suggested, the pure water flux has been added in:

Method, Line 490-491, 'The permeability of the membrane was measured using MilliQ water, demonstrating to be $218 \pm 1 \text{ L/m}^2\text{h.bar}$ (Figure S14)'.

Supporting Information,

'4.1 Pure water flux

Pure water flux of the electrochemical CNT membrane was measured over the transmembrane pressure, as shown in Figure S14.

Figure S14. Pure water flux for the fresh electrochemical CNT membrane.

The pure water flux of the CNT membrane across the transmembrane pressure ranging from 1 to 20 bar demonstrated a perfect linear relationship ($R^2=1$).

Permeability of the membrane was determined to be $218 \pm 1 \text{ L/m}^2\text{h.bar}$ as the slope of the linear fit of water flux vs. transmembrane pressure'.

Reviewer#2:

The manuscript by Liu et al. describes a method to differentiate adsorption and degradation of contaminants on an electrochemical membrane. By using HPLC-FSA and LSC together, the authors measured the concentrations of the parent compounds and the transformation products. The measurements of both parent compounds and transformation products can be used to conduct a mass balance analysis, which differentiates the adsorption and degradation. With this method, Liu et al. assessed the performance of a CNT-membrane under different operating conditions, including applied potentials, water fluxes, and contaminant concentrations.

The experiments are comprehensive, and the data analysis is sound. The authors provided detailed analysis on error estimation regarding their mass balance. I recommend acceptance of the manuscript after the authors address the following comments.

Thank you for reviewing our work and the positive assessment.

Comment 1: The authors consider adsorption/desorption and reactions as potential rate-limiting steps. However, for electrochemical reactive membranes, the rate-limiting step could be the transport of contaminants to the active sites, which motivates the development of flow-through electrodes. Please comment on the importance of adsorption/desorption versus mass transport of contaminants.

We thank the reviewer for the comment. It is true that the mass transport of contaminants could be the rate-limiting step in any electrochemical system. Therefore, we highlighted the importance of mass transport in the Introduction, Lines 42-44, *'Despite these advancements and the tremendous efforts in the development of various electrocatalysts, EO's full potential often remains underutilized due to the mass transfer limitations of reactants to the electrode surface*^{15, 16'}.

***The mass transfer can be significantly promoted by the utilization of flow-through EMR, as explained in Line 58-67, 'To address this challenge, recent research breakthroughs have led to the innovation of electrochemical membrane reactors (EMRs) that employ a conducting membrane as a flow-through electrode*^{7, 17-22}. *This setup benefits from the presence of externally modulated electrochemical potential across the membrane and in its immediate vicinity, thereby facilitating the simultaneous execution of membrane separation and electrochemical treatment processes. A critical feature of EMRs is the incorporation of nano- to micro-scale networks within the electrode, where the electrochemical reactions are spatially confined within the pores of the EMRs*²³⁻²⁵. *Such a configuration markedly reduces the thickness of the diffusion layer and increases the local concentration of reactants when compared to traditional plate electrodes, resulting in significantly improved mass transfer*^{26-28'}.**

Thus, we think that, in an EMR, the mass transfer is no longer the most significant limiting factor, which could be verified by the experiment without pre-adsorption (Figure 4B, see below). Pre-adsorption could facilitate to concentrate the micropollutant on the membrane surface prior to initiating the electrochemical reactions (if the mass transfer limitation existed). Figure 4B shows that the EMR without pre-adsorption exhibits E2 removal as high as that with pre-adsorption. This suggested that the mass transfer limitation was eliminated in the flow-through EMR, as described in Lines 240-246, *'These findings indicated that pre-adsorption of E2 did not significantly impact, either positively or negatively, the subsequent removal process. This can be attributed to; i) the apparent adsorption rate not being a limiting factor for E2 degradation in the CNT EMR under these operational conditions, owing to the high adsorption capacity of the CNT membrane and the elimination of mass transfer limitations within EMR, and ii) the desorption rate of the byproducts of E2 being rapid enough, which did not hinder the continuous adsorption of incoming E2'*.

Figure 4. (B) normalized concentration of E2 ($c_{p,E2}/c_{f,E2}$), byproduct-3m, byproduct-8m, the sum of E2 and the two byproducts, and normalized total ^3H activity over permeate volume from 0 to 500 mL. $c_{f,E2} = 1 \mu\text{g/L}$, $V_{\text{cell}} = 1.6 \text{ V}$, $J_f = 150 \text{ L/m}^2\text{h}$ (5 mL/min), 1 mM NaHCO_3 , 10 mM NaCl , 27.2 mg/L EtOH , 79.2 mg/L MeOH , pH 8.2 ± 0.2 , $23 \pm 1 \text{ }^\circ\text{C}$.

As the reviewer suggested, an additional comment concerning the importance of adsorption/desorption vs. mass transfer in the EMR has been added in the Introduction, Line 56-58, '*EMs transcend traditional membrane functions, extending beyond pure separation to embrace various electro-based strategies via several mechanisms (Error! Reference source not found.): i) mass transport of micropollutant to the membrane surface,'*, and Line 63-65, '*Given the significant alleviation of mass transfer limitations through the use of the EMR, the roles of adsorption/desorption and electron transfer processes have become increasingly crucial in the degradation of micropollutants'*'.

Comment 2: Based on the method proposed by the authors, the fraction of contaminants being removed versus being adsorbed should be a function of time (as shown by the authors in Figure 2). Over extended continuous operation, the contribution from sorption should be minimal in comparison to degradation. The authors should provide information on how time is being considered in their analysis and how the results are meaningful in terms of long-term and realistic operation.

We thank the reviewer for the comment. However, we are not completely certain about the concern of the reviewer here. We assume that the reviewer wants us to explain how the continuous long-term operation will affect the process of adsorption vs. degradation.

The reviewer is correct that the fraction of the micropollutant being removed vs. being adsorbed varies as a function of time and this was monitored. The fraction of adsorbed mass decreased consistently with time upon activating the voltage. After 160 min (corresponding to 800 mL of permeate volume), the normalized ^3H activity and E2 concentration stabilized at 1 and 0, respectively (see Figure 2). This implied that an equilibrium was reached after 800 mL of permeate, where all incoming E2 molecules were degraded and the formed by-products moved into the permeate. However, the mass balance analysis over the 1 L filtration volume demonstrated a $15 \pm 3\%$ of the initial E2 remained adsorbed on the membrane. We assumed that this was caused by the E2 adsorbed onto the inactive sites on the CNT surface, as explained in the manuscript Line 301-304, '*CNTs lack a perfect surface with sp^2 -hybridized structure and are characterized by numerous defects⁵⁷. Previous research has indicated that electroactive sites are predominantly located at these defects⁷⁴⁻⁷⁶. Consequently, the E2 molecules might adsorb onto the less electroactive sites, potentially resulting in slower rates or an inability for subsequent degradation'*'.

Figure 2. Adsorption and electrochemical degradation of E2 expressed as normalized (A) total tritium (^3H) activity ($c_{p,3\text{H}}/c_{f,3\text{H}}$) measured via LSC, and (B) concentration of E2 ($c_{p,E2}/c_{f,E2}$) measured via UHPLC-FSA in permeate vs. accumulated permeate volume. $c_{f,E2} = 1 \mu\text{g/L}$, $V_{\text{cell}} = 1.6 \text{ V}$, $J_f = 150 \text{ L/m}^2\text{h}$ (5 mL/min), 1 mM NaHCO_3 , 10 mM NaCl , 27.2 mg/L EtOH , 79.2 mg/L MeOH , pH 8.2 ± 0.2 , $23 \pm 1 \text{ }^\circ\text{C}$.

To clarify how time (permeate volume) is being considered in the analysis, an additional explanation has been added in the manuscript:

Results and Discussion, Line 162-177,

‘The mass balance analysis conducted across different permeate volume ranges revealed a clear trend: the mass removal by adsorption tended to stabilize in comparison to the increase in degradation as permeate volume increased (Figure S1). Upon activating the voltage at 500 mL of permeate, the total removed mass of E2 increased from $63 \pm 1\%$ to $76 \pm 10\%$ as the permeate volume rose from 500 to 1000 mL. Within this, the contribution from degradation consistently increased from 0 to $61 \pm 7\%$. The adsorption contribution decreased from $63 \pm 1\%$ to $19 \pm 3\%$ with the permeate volume increase from 500 to 800 mL and then plateaued in the range of 15-19% upon further increasing the permeate volume to 1000 mL. Notably, after 800 mL of filtration volume, both the normalized ^3H activity and E2 concentration stabilized at 1 and 0, respectively (Figure 2), indicating an equilibrium state where all incoming E2 molecules were degraded and the formed byproducts penetrated the permeate. However, a small fraction of E2 remained adsorbed on the membrane after 800 mL of permeate volume, suggesting some sites on the CNT surface were ineffective or unable to initiate the electrochemical reactions. The possible reasons will be further discussed in the subsequent section. These results suggested that an increase in total mass removal is anticipated with extended continuous operation (not accounting for the stability of the membrane), though a small portion of the micropollutants may remain adsorbed on the membrane’.

Supporting Information,

‘1. Contribution of adsorption and degradation to electrochemical removal of estradiol To study the electrochemical removal process as a function of the filtration volume, the mass balance analysis was conducted for the experiment at standard conditions within different permeate volume range (Figure S1).

Figure S1. Contribution of electrochemical adsorption and degradation to the mass removal of E2 within the CNT EMR as a function of the permeate volume. $c_{f,E2} = 1 \mu\text{g/L}$, $V_{cell} = 1.6 \text{ V}$, $J_f = 150 \text{ L/m}^2\text{h}$ (5 mL/min), 1 mM NaHCO_3 , 10 mM NaCl , 27.2 mg/L EtOH , 79.2 mg/L MeOH , pH 8.2 ± 0.2 , $23 \pm 1 \text{ }^\circ\text{C}$. Voltage on at 500 mL.

At 500 mL of permeate, the total mass of removed E2 was $63 \pm 1\%$, solely attributed to adsorption. Upon activating the voltage at 500 mL, total mass removal increased from $63 \pm 1\%$ to $76 \pm 10\%$ as the permeate volume increased from 500 to 1000 mL.

The contribution from degradation to total mass removal grew from 0 to $61 \pm 7\%$ over the same volume range.

Meanwhile, the contribution from adsorption decreased from $63 \pm 1\%$ to $19 \pm 3\%$ as the permeate volume increased from 500 to 800 mL and then stabilized upon further increasing the volume to 1000 mL'.

Comment 3: The authors performed their experiments using CNT electrode which has a high adsorption capacity for contaminants. To make the conclusions more universal, the authors are recommended to discuss further about other electrode materials, which may have lower adsorption capacity than CNT.

See the response to the comment 1 for reviewer#1.

Comment 4: Line 150: Please provide more explanations on why E2 can possibly get electrochemically desorbed after applying potentials. The E2 should not be charged under neutral pH, assuming NaHCO_3 was used as a buffer to maintain circumneutral pH conditions.

Indeed, the desorption of E2 retained on the CNT surface can be caused by the competition on the part of anionic species, such as Cl^- and HCO_3^- in the electrolyte, whose adsorption increases dramatically when the surface of the electrode acquires a positive charge. This is a well-known phenomenon documented for many types of electrodes [27]. Therefore, we assume that the electrochemically-induced desorption of E2 was a possible pathway in the EMR. To make it clearer, the text in the manuscript has been modified: Line 139-142, 'This increase could be caused by the electrochemical desorption of E2 that was pre-adsorbed onto the membrane due to the competition on the part of anionic species⁶⁶, such as Cl^- and HCO_3^- present in the electrolyte'.

Comment 5: Figure 3B: It's unclear how the authors normalized the permeate concentrations of the transformation products by the feed concentrations of the transformation products. The feed concentrations of the transformation products should be zero.

We appreciate the reviewer's thoughtful observation. The permeate concentrations of the byproducts were normalized by the initial feed concentration of E2. To clarify this, an explanation has been added in the Methods Section, Line 570-576, *'To compare the formation of degraded products, the permeate concentrations were normalized by the initial SH concentration in the feed ($c_{p,prod}/c_{f,SH}$). As the identification of the byproducts has not been possible, accurately determining their concentrations using calibration was not feasible. Therefore, the concentration of the byproducts was estimated using the calibration for the parent SH. It is important to note that these concentrations, as defined in this study, do not represent the actual concentrations of each byproduct, but are used for quantitative comparisons of the formation of the same byproduct at varying conditions'*.

Comment 6: Line 158-162, the authors claimed this is an effective technology for treatment. However, high concentrations of oxidation products are released as evidenced in Figure 3 and 4. The authors should comment on the inability of this system in mineralizing contaminants.

Thank the reviewer for the comment. It is true that a high concentration of byproducts was formed. As the reviewer suggested, a description has been added in the manuscript in Lines 179-182, *'While the CNT EMR process demonstrated effective degradation of the target micropollutant, complete mineralization could not be achieved under the conditions tested. This partial degradation, leading to the formation of byproducts, identified by the UHPLC-FSA chromatogram analysis (Figure 3)'*.

It has been reported in the literature that the electrocatalysis via direct oxidation tends to reduce the estrogenic activity of SH, as described in Line 198-199, *'Prior studies⁶⁹⁻⁷¹ have identified a ketone derivative as a product of the direct electrochemical oxidation of E2 through a two-electron transfer mechanism, as evidenced by GC-MS analysis. Such a transformation was anticipated to weaken the estrogenic activity of E2. This effect was primarily because of the conversion of the phenol group, a crucial structural component that enables endocrine disrupting chemicals to exhibit their estrogenic properties through interaction with the human estrogen receptor, into a quinone-like moiety^{70, 72, 73'}. Additionally, it is important to further study the estrogenic activity of the treated water, and a description has been added in Lines 203-204, *'Further investigation into the estrogenic activity of the EMR-treated SH water is crucial and is currently in progress'*.*

Comment 7: Figure 4C: For the analysis with pre-adsorption, the authors should provide explanations on the rationale of taking the two "in permeate (unremoved)" as two independent pathways. They are basically the residual after adsorption and reaction, why this unremoved part under no potential and the one with potential can be taken as individual process and can be stacked together for mass balance? Is this because the integration of the area under the curve for these two masses are conducted in separate time periods? If so, please clarify the manuscript.

We thank the reviewer for the comment. Exactly, the two 'in permeate (unremoved)' considering as two individual processes because the integration of the area was conducted for separate time periods. As the reviewer suggested, an explanation of the mass balance analysis has been added in the main text and SI, as shown below.

Line 235-238, *'Additionally, after turning on the voltage, 5±2% of the E2 was released into the permeate, which was comparable to the 2±2% of E2 that remained unremoved without pre-adsorption. Details of the mass balance analysis for the experiments with and without pre-adsorption are available in Figure S22'*.

Supporting Information,

'5.6 Mass balance analysis for the pre-adsorption experiments

As an illustration of the mass balance analysis conducted using the integrated UHPLC-FSA and LSC method, the contribution of degraded, adsorbed, and unremoved E2 in the experiments with and without pre-adsorption, is shown in Figure S22.

Figure S22. Illustration of the mass balance analysis of the degraded, adsorbed, and unremoved E2 in the experiments with and without pre-adsorption.

In the pre-adsorption experiment (Figure S22A), prior to activating the voltage, $19\pm4\%$ of the initial E2 mass was released into the permeate (unremoved), corresponding to the integrated area ① over the volume range of 0-500 mL. Additionally, $31\pm4\%$ was adsorbed onto the membrane (area ②). After activating the voltage, $5\pm2\%$ of the E2 passed through the system unremoved, as determined by the integrated area ⑤ for the 500-1000 mL permeate volume. In total, $61\pm7\%$ of E2 was degraded (area ③+④) during the degradation phase (500-1000 mL), while $15\pm3\%$ remained adsorbed on the membrane (area ②-④).

In the experiment without pre-adsorption (Figure S22B), $77\pm3\%$ of E2 was degraded (area ①) and $22\pm4\%$ (area ②) was adsorbed onto the membrane, while the remaining $2\pm2\%$ was released into the permeate'.

Comment 8: Figure 4C, why is the in permeate (power off) and adsorbed on membrane prior to applying voltage adds up to 50%? Also, can the authors please check if the bottom right label "in permeate (unremoved)" is labelled correctly with the arrow pointing to the blue bar?

In the experiment with pre-adsorption, 500 mL of the feed was first filtered with the power off, followed by another 500 mL of filtration with the power on, as described in the Method Section Line 475-477 'The membrane was pre-adsorbed with 500 mL of SHs feed solution without imposing a cell voltage. Afterwards, the electric power was activated, and another 500 mL of feed was subjected to treatment'. Consequently, the E2 mass that was filtered prior to applying voltage, which ended up either in the permeate or adsorbed on the membrane, accounted for 50% of the initial mass in the feed.

Regarding the label 'in permeate (unremoved)', it was incorrectly labeled due to an oversight and has been corrected to 'Degraded' (as shown below). We thank the reviewer for pointing to this.

Figure 4 (C) contribution to the mass removal of E2 by the electrochemical adsorption and degradation. $c_{f,E2} = 1 \mu\text{g/L}$, $V_{cell} = 1.6 \text{ V}$, $J_f = 150 \text{ L/m}^2\text{h}$ (5 mL/min), 1 mM NaHCO_3 , 10 mM NaCl , 27.2 mg/L EtOH, 79.2 mg/L MeOH, pH 8.2 ± 0.2 , $23 \pm 1 \text{ }^\circ\text{C}$.

Comment 9: Line 1107: The authors should justify their assumptions that all byproducts are fully desorbed. This assumption is crucial for the analysis of mass balance. However, there is no experimental evidence (e.g., extraction) to confirm this assumption.

We thank the reviewer for the comment. As shown in Figure 2, after the pre-adsorbed E2 was degraded and desorbed from the membrane, an equilibrium was reached where the normalized ^3H and E2 stabilized at 1 and 0, respectively. This suggested that all amount of the incoming E2 was degraded (normalized E2 was 0) and the byproducts are desorbed (normalized ^3H was 1).

Figure 2. Adsorption and electrochemical degradation of E2 expressed as normalized (A) total tritium (^3H) activity ($c_{p,3H}/c_{f,3H}$) measured via LSC, and (B) concentration of E2 ($c_{p,E2}/c_{f,E2}$) measured via UHPLC-FSA in permeate vs. accumulated permeate volume. $c_{f,E2} = 1 \mu\text{g/L}$, $V_{cell} = 1.6 \text{ V}$, $J_f = 150 \text{ L/m}^2\text{h}$ (5 mL/min), 1 mM NaHCO_3 , 10 mM NaCl , 27.2 mg/L EtOH, 79.2 mg/L MeOH, pH 8.2 ± 0.2 , $23 \pm 1 \text{ }^\circ\text{C}$.

Comment 10: Figure 7, can the authors provide explanations on why there was no sorption for the 10^5 ng/L experiment?

Indeed, the adsorbed mass at a concentration of 10^5 ng/L was significant. The mass of E2 adsorbed onto the membrane, as quantified using the method outlined in Method Section 4.6, was 9.4 ± 1.6 , 10.6 ± 3 , 150 ± 32 , 996 ± 312 , 2155 ± 3340 , 8077 ± 17625 , and 47088 ± 33796 ng at E2 concentrations of 50, 100, 10^3 , 10^4 , 10^5 , $5 \cdot 10^5$, and 10^6 ng/L, respectively, showing an increase with rising E2 concentrations. However, at 10^5 ng/L, the adsorbed E2 mass accounted for only $2.2 \pm 3.4\%$ of the initial E2 mass (98682 ± 3626 ng), which appears negligible on the bar graph.

To make this clearer, an explanation of the sorption has been added in the manuscript Lines 376-379: *'The adsorption contribution declined sharply at concentrations higher than 10^4 ng/L. It is worth noting that the mass of adsorbed E2 consistently increased with the rising E2 concentrations. However, due to the large initial mass of E2, the proportion of adsorbed E2 appears negligible at concentrations $> 10^4$ ng/L.'*

Comment 11: Line 99, the sentence is not complete. "Formation of active (missing word) in electrochemical processes".

Thank the reviewer for pointing to this. The missing word 'species' has been added and the complete sentence is 'The ubiquity of chloride ions (Cl^-) in natural waters and wastewater effluents often leads to the formation of active species in electrochemical processes'.

References

- [1] K.L. Thorpe, T.H. Hutchinson, M.J. Hetheridge, M. Scholze, J.P. Sumpter, C.R. Tyler, Assessing the biological potency of binary Mixtures of environmental estrogens using vitellogenin induction in juvenile rainbow trout (*oncorhynchus mykiss*), *Environmental Science & Technology*, 35 (2001) 2476-2481
- [2] World Health Organization & UNICEF, Billions of people will lack access to safe water, sanitation and hygiene in 2030 unless progress quadruples – warn WHO, in, World Health Organization, <https://www.who.int/news/item/01-07-2021-billions-of-people-will-lack-access-to-safe-water-sanitation-and-hygiene-in-2030-unless-progress-quadruples-warn-who-unicef>, 2021.
- [3] M. Sun, X. Wang, L.R. Winter, Y. Zhao, W. Ma, T. Hedtke, J.-H. Kim, M. Elimelech, Electrified membranes for water treatment applications, *ACS ES&T Engineering*, (2021)
- [4] G.d.S. Cunha, B.M.d. Souza-Chaves, D.M. Bila, J.P. Bassin, C.D. Vecitis, M. Dezotti, Insights into estrogenic activity removal using carbon nanotube electrochemical filter, *Science of The Total Environment*, 678 (2019) 448-456
- [5] Y. Liu, H. Liu, Z. Zhou, T. Wang, C.N. Ong, C.D. Vecitis, Degradation of the common aqueous antibiotic tetracycline using a carbon nanotube electrochemical filter, *Environmental Science & Technology*, 49 (2015) 7974-7980
- [6] J. Radjenovic, N. Duinslaeger, S.S. Avval, B.P. Chaplin, Facing the challenge of poly- and perfluoroalkyl substances in water: Is electrochemical oxidation the answer?, *Environmental Science & Technology*, 54 (2020) 14815-14829
- [7] M. Šimková, L. Kolátorová, P. Drašar, J. Vítků, An LC-MS/MS method for the simultaneous quantification of 32 steroids in human plasma, *Journal of Chromatography B*, 1201-1202 (2022) 123294
- [8] W. Liu, D. Yuan, M. Han, J. Huang, Y. Xie, Development and validation of a sensitive LC-MS/MS method for simultaneous quantification of thirteen steroid hormones in human serum and its application to the study of type 2 diabetes mellitus, *Journal of Pharmaceutical and Biomedical Analysis*, 199 (2021) 114059
- [9] M.M. Ngundi, O.A. Sadik, T. Yamaguchi, S.-i. Suye, First comparative reaction mechanisms of β -estradiol and selected environmental hormones in a redox environment, *Electrochemistry Communications*, 5 (2003) 61-67

- [10] Y. Ohko, K.-i. Iuchi, C. Niwa, T. Tatsuma, T. Nakashima, T. Iguchi, Y. Kubota, A. Fujishima, 17 β -estradiol degradation by TiO₂ photocatalysis as a means of reducing estrogenic activity, *Environmental Science & Technology*, 36 (2002) 4175-4181
- [11] J. Mai, W. Sun, L. Xiong, Y. Liu, J. Ni, Titanium dioxide mediated photocatalytic degradation of 17 β -estradiol in aqueous solution, *Chemosphere*, 73 (2008) 600-606
- [12] S. Gligorovski, R. Strekowski, S. Barbati, D. Vione, Environmental implications of hydroxyl radicals (\bullet OH), *Chemical Reviews*, 115 (2015) 13051-13092
- [13] H. Dodgen, H. Taube, The exchange of chlorine dioxide with chlorite ion and with chlorine in other oxidation states, *Journal of the American Chemical Society*, 71 (1949) 2501-2504
- [14] S. Neodo, D. Rosestolato, S. Ferro, A. De Battisti, On the electrolysis of dilute chloride solutions: Influence of the electrode material on Faradaic efficiency for active chlorine, chlorate and perchlorate, *Electrochimica Acta*, 80 (2012) 282-291
- [15] H.B. Ammar, M.B. Brahim, R. Abdelhédi, Y. Samet, Green electrochemical process for metronidazole degradation at BDD anode in aqueous solutions via direct and indirect oxidation, *Separation and Purification Technology*, 157 (2016) 9-16
- [16] J.D. García-Espinoza, P. Mijaylova-Nacheva, M. Avilés-Flores, Electrochemical carbamazepine degradation: Effect of the generated active chlorine, transformation pathways and toxicity, *Chemosphere*, 192 (2018) 142-151
- [17] J. Zhang, Y. Nosaka, Mechanism of the OH Radical Generation in Photocatalysis with TiO₂ of Different Crystalline Types, *The Journal of Physical Chemistry C*, 118 (2014) 10824-10832
- [18] Y. Nosaka, A.Y. Nosaka, Generation and detection of reactive oxygen species in photocatalysis, *Chemical Reviews*, 117 (2017) 11302-11336
- [19] J. Zhang, Y. Nosaka, Quantitative Detection of OH Radicals for Investigating the Reaction Mechanism of Various Visible-Light TiO₂ Photocatalysts in Aqueous Suspension, *The Journal of Physical Chemistry C*, 117 (2013) 1383-1391
- [20] K. Fischer, P. Schulz, I. Atanasov, A. Abdul Latif, I. Thomas, M. Kühnert, A. Prager, J. Griebel, A. Schulze, Synthesis of high crystalline TiO₂ nanoparticles on a polymer membrane to degrade pollutants from water, *Catalysts*, 8 (2018)
- [21] C.A. Martínez-Huitle, M.A. Rodrigo, I. Sirés, O. Scialdone, Single and coupled electrochemical processes and reactors for the abatement of organic water pollutants: A critical review, *Chemical Reviews*, 115 (2015) 13362-13407
- [22] C. Comninellis, Electrocatalysis in the electrochemical conversion/combustion of organic pollutants for waste water treatment, *Electrochimica Acta*, 39 (1994) 1857-1862
- [23] B. Marselli, J. Garcia-Gomez, P.A. Michaud, M.A. Rodrigo, C. Comninellis, Electrogeneration of Hydroxyl Radicals on Boron-Doped Diamond Electrodes, *Journal of The Electrochemical Society*, 150 (2003) D79
- [24] W. Duan, G. Chen, C. Chen, R. Sanghvi, A. Iddya, S. Walker, H. Liu, A. Ronen, D. Jassby, Electrochemical removal of hexavalent chromium using electrically conducting carbon nanotube/polymer composite ultrafiltration membranes, *Journal of Membrane Science*, 531 (2017) 160-171
- [25] W. Duan, A. Ronen, S. Walker, D. Jassby, Polyaniline-coated carbon nanotube ultrafiltration membranes: Enhanced anodic stability for in situ cleaning and electro-oxidation processes, *ACS Applied Materials & Interfaces*, 8 (2016) 22574-22584
- [26] A.V. Dudchenko, J. Rolf, K. Russell, W. Duan, D. Jassby, Organic fouling inhibition on electrically conducting carbon nanotube-polyvinyl alcohol composite ultrafiltration membranes, *Journal of Membrane Science*, 468 (2014) 1-10
- [27] H.J. Kreuzer, Kinetics of Adsorption, Desorption and Reactions at Surfaces, in: M. Rocca, T.S. Rahman, L. Vattuone (Eds.) *Springer Handbook of Surface Science*, Springer International Publishing, Cham, 2020, pp. 1035-1052.

REVIEWER COMMENTS

Reviewer #1 (Remarks to the Author):

The revised manuscript is now in a good shape. There are some issues remained, which need further clarifications before acceptance.

1. I am satisfied with the author's revisions except for the Comment 1 of Reviewer #1. I insist that it is important to investigate this coupled analysis method on other electrodes with lower adsorption capacity, such as TiO₂ or Magnéli phase Ti₄O₇, etc. Reviewer #2 (Comment 3) also pointed out this issue, indicating the importance of this issue. Although the authors stated that the primary objective of this study was to address the removal of micropollutants from water by membrane electrocatalysis, I think that the analytical method is the main innovation worth publishing. As I know, the authors have reported membrane photocatalysis for the removal of SHs (Nat. Nanotechnol. 17, 417–423 (2022)). This work has only altered a degradation process in the reported membrane materials and electrochemical devices. It is therefore extremely important to demonstrate the universality of the analytical method.

2. This manuscript include too many references. Is it possible to remove some of the unnecessary ones?

Reviewer #2 (Remarks to the Author):

The revisions are detailed and adequate. I recommend acceptance.

We greatly appreciate the reviewers' comments. Specific responses to the reviewer's comments are provided below, and corresponding changes that have been made in the manuscript are highlighted in this revision.

COMMENTS FROM THE REVIEWERS

Reviewer#1:

The revised manuscript is now in a good shape. There are some issues remained, which need further clarifications before acceptance.

Comment 1: I am satisfied with the author's revisions except for the Comment 1 of Reviewer #1. I insist that it is important to investigate this coupled analysis method on other electrodes with lower adsorption capacity, such as TiO₂ or Magnéli phase Ti₄O₇, etc. Reviewer #2 (Comment 3) also pointed out this issue, indicating the importance of this issue. Although the authors stated that the primary objective of this study was to address the removal of micropollutants from water by membrane electrocatalysis, I think that the analytical method is the main innovation worth publishing. As I know, the authors have reported membrane photocatalysis for the removal of SHs (Nat. Nanotechnol. 17, 417–423 (2022)). This work has only altered a degradation process in the reported membrane materials and electrochemical devices. It is therefore extremely important to demonstrate the universality of the analytical method.

Thank the reviewer again for the thoughtful comment and suggestion.

The reviewer's request to investigate other electrode materials like TiO₂ or Magnéli phase Ti₄O₇ is appreciated. However, TiO₂ is a semiconductor and generally this material is unsuitable for direct use in electrochemical oxidation process. Only few studies reported the use of conductive TiO₂ nanotubes formed on Ti substrate using electrochemical anodization method [1]. Additionally, the fabrication of Ti₄O₇ electrode membrane is complex, involving the treatment under 1 atm H₂ and 1050 °C for 10 h to convert TiO₂ to Ti₄O₇ [2]. These fabrication challenges and the specialized preparation required for TiO₂ nanotubes and Ti₄O₇ significantly exceed the stated scope of our current paper.

To address the reviewer's concern regarding the applicability of our analysis tools for investigating materials with lower adsorption capacity, we have indeed applied this analytical method to a membrane photocatalysis process in a parallel study (the manuscript with relevant results is to be submitted soon). In that study, a PVDF-TiO₂ membrane that exhibits a minimal adsorption capacity, was utilized to treat estradiol (E2) micropollutant in a flow-through system [3] under solar irradiation using two membrane fluxes, 60 and 3000 L/m²h (some representative results are shown in Figure R1 below).

Figure R1. Photocatalytic degradation of E2 using a PVDF-TiO₂ membrane under solar irradiation, as normalized concentration of E2, and ³H in permeate vs. accumulated permeate volume at flux (A) 60 and (B) 3000 L/m²h. $C_{f,E2} = 100$ ng/L, $I = 50$ mW/cm², pH 8.3 ± 0.2 , 23 ± 1 °C, 1 mM NaHCO₃, 10 mM NaCl, 27.2 mg/L EtOH, and 79.2 mg/L MeOH.

The results show that at both selected conditions, the adsorption of E2 onto the membrane was negligible, demonstrating a normalized ³H concentration around 1 throughout the entire filtration process. Meanwhile, the use of a lower flux significantly resulted in an enhanced the steady-state E2 removal, increasing from 0 to $82 \pm 4\%$ when the flux was reduced from 3000 to 60 L/m²h). In both cases, the contributions of the adsorption and photocatalytic degradation were distinguishable using the developed analysis approach (Figure R2), suggesting a minimal contribution from adsorption at both 60 and 3000 L/m²h, and a significantly higher contribution from degradation at the lower flux 60 L/m²h.

Figure R2. Contribution to the mass removal of E2 by the adsorption and photocatalytic degradation using a PVDF-TiO₂ membrane under solar irradiation at flux 60 and 3000 L/m²h. $C_{f,E2} = 100$ ng/L, $I = 50$ mW/cm², pH 8.3 ± 0.2 , 23 ± 1 °C, 1 mM NaHCO₃, 10 mM NaCl, 27.2 mg/L EtOH, and 79.2 mg/L MeOH.

In fact, the model we selected for investigating the performance of the EMR for micropollutant removal is one of the most complex cases in electrochemical systems, where the processes of electrochemical adsorption, desorption, degradation, and byproduct formation occur simultaneously in the presence of at extremely low concentrations of the target pollutant. This complexity necessitates the development of an array of analytical methods capable of differentiating these processes and thus to deepen our understanding of the underlying mechanisms. In scenarios where the EMR (and/or photocatalytic membrane reactor) exhibits very low adsorption capacity or degradation efficiency, interpreting the system could be simpler. However, our developed analytical methodology remains applicable across different scenarios.

To clarify the reviewer's concern about the universality of this work, an explanation has been added in the manuscript, Line 441-445, '*These findings highlighted the innovative nature of integrating UHPLC-FSA and LSC techniques to investigate the fundamental mechanisms of the complex EMR processes that involving an intricate interplay of multiple reactions—adsorption/desorption, degradation and byproduct transformation—that occur concurrently. This method can be can be readily applied to other EMR systems, such as those without adsorption/desorption or byproduct formation*'.

Comment 2: This manuscript include too many references. Is it possible to remove some of the unnecessary ones?

Thank the reviewer for pointing to this. As the reviewer suggested, some old and less relevant references have been removed. The total quantity of the reference has been reduced from 91 to 79.

Reference

- [1] S.P. Albu, A. Ghicov, S. Berger, H. Jha, P. Schmuki, TiO₂ nanotube layers: Flexible and electrically active flow-through membranes, *Electrochemistry Communications*, 12 (2010) 1352-1355
- [2] L. Hua, H. Cao, Q. Ma, X. Shi, X. Zhang, W. Zhang, Microalgae Filtration Using an Electrochemically Reactive Ceramic Membrane: Filtration Performances, Fouling Kinetics, and Foulant Layer Characteristics, *Environmental Science & Technology*, 54 (2020) 2012-2021
- [3] S. Lotfi, K. Fischer, A. Schulze, A.I. Schäfer, Photocatalytic degradation of steroid hormone micropollutants by TiO₂-coated polyethersulfone membranes in a continuous flow-through process, *Nature Nanotechnology*, 17 (2022) 417-423

REVIEWERS' COMMENTS

Reviewer #1 (Remarks to the Author):

I recommend the publication of this work as is.